

# An Improved Parameterization of Sea Spray-Mediated Heat Flux Using Gaussian Quadrature: Case Studies with a Coupled CFSv2.0-WW3 System

Ruizi Shi[1] and Fanghua Xu[1*]

[1] Department of Earth System Science, Ministry of Education Key Laboratory for Earth System Modeling, Institute for Global Change Studies, Tsinghua University, Beijing, 100084, China.

*Correspondence to*: Fanghua Xu (fxu@mail.tsinghua.edu.cn)





**Abstract.** Sea spray-mediated heat flux plays an important role in air-sea heat transfer. Heat flux
integrated over droplet size spectrum can well simulate total heat flux induced by sea spray droplets.
Previously, a fast spray-flux scheme assuming single-radius droplets (A15) was widely used since the
full-size spectrum integral is computational expensive. Based on the Gaussian Quadrature (GQ) method,
a new fast scheme (SPRAY-GQ) of sea spray-mediated heat flux is derived. The performance of SPRAY-
GQ is evaluated by comparing heat fluxes with those estimated from the widely-used A15. The new
scheme shows a better agreement with the original spectrum integral. To further evaluate the performance
of A15 and SPRAY-GQ, the two schemes are implemented into a coupled CFSv2.0-WW3 system, and
a series of 56-day simulations in summer and winter are conducted and compared. The comparisons with
satellite measurements and reanalysis data show that the SPRAY-GQ scheme could simulate air-sea heat
flux more reasonably than the A15 scheme. For experiments based on SPRAY-GQ, the sea surface
temperature at mid-high latitudes of both hemispheres, particularly in summer, is significantly improved
compared with the experiments based on A15. The simulation of 10-m wind speed and significant wave
height at mid-low latitudes of the Northern Hemisphere is improved as well. The computational time of
SPRAY-GQ is about the same as that of A15. Thereby, the newly-developed SPRAY-GQ scheme has a
potential to be used for improving air-sea heat flux in coupled models.



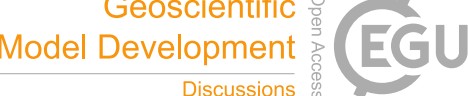
## 1 Introduction

Sea spray droplets, ejected from oceans, include film drops, jet drops and spume drops (Veron, 2015).
The first two types of droplets are generated from bubble bursting caused by ocean surface wave breaking,
with radius ranging from 0.5 μm to 50 μm (Resch and Afeti, 1991; Thorpe, 1992; Melville, 1996; Spiel,
1997; Andreas, 1998; Lhuissier and Villermaux, 2012). Spume drops are generated by strong winds (>
7-11 m/s) which directly tear the wave crests, with larger radius ranging from tens to hundreds μm
(Koga, 1981; Andreas et al., 1995; Andreas, 1998). Sea spray droplets play an important role in weather
and climate processes (Fox-Kemper et al., 2022). On one hand, sea spray droplets contribute to local
marine aerosols and subsequently modify the local radiation balance (Fairall et al., 1983; Burk, 1984;
Fairall and Larsen, 1984). On the other hand, sea spray droplets affect the fluxes of heat, momentum,
salt, and freshwater between atmosphere and ocean (Andreas, 1992; Andreas et al., 2008; Andreas, 2010;
Andreas et al., 2015; Ling and Kao, 1976; Fairall et al., 1994; Andreas and Decosmo, 2002).
The sea spray-mediated heat transfer mainly occurs within the droplet evaporation layer (DEL) near
the sea surface (Andreas and Decosmo, 1999, 2002; Fairall et al., 1994). Sea spray droplets with the same
temperature as ocean surface can lead to sensible heat flux in DEL, while water evaporated from these
droplets can further release latent heat to the atmosphere (Andreas, 1992; Borisenkov, 1974; Bortkovskii,
1973; Wu, 1974; Monahan and Van Patten, 1988; Ling and Kao, 1976). Part of the sea spray-mediated
sensible heat is absorbed by droplet evaporation, which further increases the air-sea temperature
difference, and thus increases the sea spray-mediated sensible heat flux (Fairall et al., 1994; Andreas and
Decosmo, 2002). Since strong winds produce more sea spray droplets with larger radius, sea spray-
mediated heat fluxes increase with wind speed (Fairall et al., 1994), and contribute more than 10% of the
total surface heat flux after reaching the threshold speed (> 11 m/s for sensible heat flux and > 13 m/s
for latent heat flux)(Andreas et al., 2008). In addition, when a droplet is released into the air, it is
accelerated due to surface winds (Edson and Andreas, 1997; Fairall et al., 1994; Van Eijk et al., 2011;
Wu et al., 2017). If the droplet could fall back into the ocean, additional momentum would be injected
into the ocean from the atmosphere (Andreas, 1992, 2004).
The usual bulk parameterizations in numerical models for surface fluxes only include the interfacial
(turbulent) fluxes (e.g., Fairall et al., 1996), while neglecting the significant contributions of sea spray





droplets in DEL (Andreas et al., 2008; Fairall et al., 1994; Smith, 1997; Emanuel, 1995). Andreas and
Emanuel (2001) implemented sea spray-mediated heat flux and momentum flux parameterizations into
a simple tropical cyclone model, and found that the sea spray-induced heat flux significantly enhances
the tropical cyclone intensity, offsetting the negative effect of enhanced surface drag by strong wind and
waves. The similar enhancement of tropical cyclone intensity was also shown in recent regional coupling
systems by including sea spray-mediated heat flux (Xu et al., 2021a; Liu et al., 2012; Garg et al., 2018;
Zhao et al., 2017). In the First Institute of Oceanography Earth System Model, Bao et al. (2020) first
incorporated the sea spray-mediated heat flux in global climate simulation. Following Bao et al. (2020),
Song et al. (2022) found the sea spray-mediated heat flux can lead to cooling at the air-sea interface and
strengthening westerlies in the Southern Ocean, and thus improves estimates of sea surface temperature
(SST).

Since the parameterization of sea spray-mediated heat flux derived from observations requires full-

size spectral integral and demands huge amount of computational time (Andreas, 1989, 1990, 1992;
Andreas et al., 2015), a simplified parameterization based on a single radius of sea spray droplets
(Andreas et al., 2015; Andreas et al., 2008) is widely used in atmosphere-ocean coupling systems (Xu et
al., 2021a; Liu et al., 2012; Garg et al., 2018; Zhao et al., 2017; Song et al., 2022; Bao et al., 2020), and
apt to produce significant biases. To reduce biases induced by the single radius of sea spray droplets, we
develop a new parameterization of sea spray-mediated heat flux based on the Gaussian Quadrature (GQ)
method, a fast and accurate way to calculate spectral integral. The GQ method has been successfully
used for the estimation of domain-averaged radiative flux profiles (Li and Barker, 2018). The
performance of the GQ-based parameterization of the sea spray-mediated heat flux is evaluated and
compared with the simplified parameterization for single radius of Andreas et al. (2015), referred to as
A15 hereafter. The results are first compared with the original parameterization using full-size spectral
integral (A92, hereafter). Then the parameterizations are implemented in a global coupled atmosphere-
ocean-wave system (Shi et al., 2022), and the results are compared with global satellite measurements
and reanalysis data.

The rest of the paper is structured as follows: observation and reanalysis data for comparisons are

introduced in Section 2; the derivation of the GQ-based parameterization and the global coupling system





are described in Section 3; the performance of the new parameterization is evaluated in Section 4. Finally,
a summary and discussion are given in Section 5.
**2 Data**

The fifth generation European Centre for Medium-Range Weather Forecasts (ECMWF) Reanalysis

(ERA5; Hersbach et al., 2020) data assimilated huge amounts of historical data and thus provided reliable
hourly estimates. ERA5 10-m wind speed (WSP10), 2-m air temperature, 2-m dewpoint temperature,
surface pressure and significant wave height (SWH) with a spatial resolution of 0.5° are used.
Additionally, WSP10, 2-m air temperature and 2-m specific humidity data from the Objectively
Analyzed air-sea Fluxes (OAFlux) products (Yu et al., 2008) are also applied for comparison, with 1°×1°
resolution. The daily average satellite Optimum Interpolation SST (OISST) data are obtained from the
National Oceanic and Atmospheric Administration (NOAA) with a spatial resolution of 0.25° (Reynolds
et al., 2007). The global monthly mean salinity observation from European Space Agency (ESA;
https://climate.esa.int/sites/default/files/SSS_cci-D1.1-URD-v1r4_signed-accepted.pdf ) are applied.
**3 Methods**
**3.1 Development of a Fast Algorithm Based on GQ**

The effects of sea spray droplets on sensible and latent heat fluxes ($H_{S,SP}$, $H_{L,SP}$) contribute to the total

sensible and latent heat fluxes ($H_{S,T}$, $H_{L,T}$) at the air-sea interface. That is,

$$H_{S,T} = H_S + H_{S,SP}, \qquad (1)$$

$$H_{L,T} = H_L + H_{L,SP}. \qquad (2)$$

where $H_S$ and $H_L$ are the sensible and latent heat fluxes at the air-sea interface due to the air-sea
differences of temperature and humidity. Based on eddy correlation observations, A92 (Andreas, 1989,
1990, 1992; Andreas et al., 2015) integrates the sea spray-mediated sensible and latent heat flux
spectrums over initial droplet radius ($Q_S(r_0)$ and $Q_L(r_0)$) to estimate $H_{S,SP}$ and $H_{L,SP}$ (details in
Appendix A). The distributions of $Q_S(r_0)$ and $Q_L(r_0)$ spectrums as functions of initial droplet radius
$r_0$ under various atmosphere and ocean state are shown in Fig. 1, indicating that $Q_S$ and $Q_L$ spectrums





are more sensitive to the change of 10-m wind speed, and less sensitive to other variables, including 2-
m air temperature, 2-m relative humidity, sea surface temperature, surface air pressure and sea surface
salinity.

The calculation of $H_{S,SP}$ and $H_{L,SP}$ in A92 requires huge amount of computational time due to full-

size spectral integral (Eqn. A5-A6 of Appendix A), therefore it is difficult to apply A92 directly in
coupled modeling systems. A15 (Andreas et al., 2015) developed a fast algorithm by using a single
representative droplet radius (details in Appendix B), which was widely adopted in recent reginal and
global coupling systems (Xu et al., 2021a; Liu et al., 2012; Garg et al., 2018; Zhao et al., 2017; Song et
al., 2022; Bao et al., 2020). In this study, we apply a 3-node GQ method (details in Appendix C) to
develop a new fast algorithm to approximate the full-size spectral integral of A92. Notably, GQ can
converge exponentially to the actual integral only for a smooth function (McClarren, 2018). Since as
functions of $r_0$, $Q_S$ and $Q_L$ are not smooth (Fig. 1), a data sorting from largest to smallest is required.
After sorting, $Q_S$ and $Q_L$ become $Q_{SS}$ and $Q_{LS}$, and then GQ can be used to estimate the integral of
$Q_{SS}$ and $Q_{LS}$. However, the sorting leads to high complexity of GQ comparable to A92. Therefore, it is
necessary to find the general law of GQ nodes for $Q_{SS}$ and $Q_{LS}$ to avoid the sorting in application.

To derive the general law of GQ nodes, we calculate the distribution of the sea spray-mediated heat

flux spectral following A92, based on the global daily WSP10, 2-m air temperature, 2-m dewpoint
temperature, surface pressure and SWH of ERA5 and OISST from August 1, 2018 to August 31, 2018.
The ESA monthly salinity is also applied since the sea spray-mediated heat flux is the least sensitive to
salinity (Fig. 1e&f) and only monthly salinity observation data is available. From the global spectrums,
we sort $Q_S$ and $Q_L$ from largest to smallest. The GQ nodes corresponding to $r_0$ of the sensible (latent)
heat flux after sorting are denoted as $r_{S1}$, $r_{S2}$ and $r_{S3}$ ($r_{L1}$, $r_{L2}$ and $r_{L3}$), whose distribution of
occurrence frequency in percentage is shown in Fig. 2. It is noted that except that $r_{L3}$ is related to
WSP10 (Fig. 2c), all other five nodes have frequency roughly concentrated at a constant (peak
frequency >65%), that is

$$r_{S1} = 459.056, \; r_{S2} = 294.185, \; r_{S3} = 166.771, \qquad (3)$$

$$r_{L1} = 443.914, r_{L2} = 251.0498, \qquad (4)$$



$$r_{L3} = \begin{cases} 60.310 \, WSP10^{0.1161}, & WSP10 \geq 2 \, m/s \\ 58.086, & WSP10 < 2 \, m/s \end{cases}, \qquad (5)$$

where the unit of the radius is micrometer. And then the 3-node GQ to approximate the full-size spectral
integral of A92 are

$$\int_a^b Q_S(r_0) dr_0 \approx \frac{b-a}{2} \sum_{i=1}^3 \omega_i Q_S(r_{Si}), \qquad (6)$$

$$\int_a^b Q_L(r_0) dr_0 \approx \frac{b-a}{2} \sum_{i=1}^3 \omega_i Q_L(r_{Li}). \qquad (7)$$

Here a and b are the lower and upper limits of $r_0$, which are set to $2 \mu m$ and $500 \mu m$ based on Andreas
(1990), and $\omega_i$ is the corresponding weight ($\omega_1 = \omega_3 = 0.556$, $\omega_2 = 0.889$), obtained from McClarren
(2018). Thus, we can directly use Eqn. (3-7) to estimate the GQ-based $H_{S,SP}$ and $H_{L,SP}$ approximations,
avoiding sorting. The new fast algorithm is referred to as SPRAY-GQ hereafter.

**3.2 CFSv2.0-WW3 Coupling System**

A coupled system based on Climate Forecast System model version 2.0 (CFSv2.0) and
WAVEWATCH III (WW3) is employed to evaluate and compare the effects of sea spray-mediated heat
flux parameterized by A15 and SPRAY-GQ. The CFSv2.0-WW3 has three components, the Global
Forecast System (GFS; http://www.emc.ncep.noaa.gov/GFS/doc.php) as the atmosphere component of
CFSv2.0, the Modular Ocean Model version 4 (MOM4; Griffies et al., 2004) as the ocean component of
CFSv2.0, and the WW3 (WAVEWATCH III Development Group, 2016) as the ocean surface wave
component. The variables between CFSv2.0 and WW3 are interpolated and passed using the Chinese
Community Coupler version 2.0 (C-Coupler2; Liu et al., 2018).
The CFSv2.0 is mainly applied for intraseasonal and seasonal prediction (e.g., Saha et al., 2014). The
atmosphere component GFS uses a spectral triangular truncation of 382 waves (T382) in the horizontal,
equivalent to a grid resolution of nearly 35 km, and 64 sigma-pressure hybrid layers in the vertical. The
MOM4 is integrated on a nominal 0.5° horizontal grid with enhanced horizontal resolution to 0.25° in
the tropics, and there are 40 levels in the vertical. The latitude range of WW3 is 78°S–78°N with a spatial
resolution of 1/3°. In the coupling system, the WW3 obtains 10-m wind and ocean surface current from
CFSv2.0, and then provides wave parameters to CFSv2.0. Several wave-mediated processes, including



upper ocean mixing modified by Stokes drift-related processes, air-sea fluxes modified by surface current
and Stokes drift, and momentum roughness length, are considered. Details of this system are referred to
Shi et al. (2022).

A series of numerical experiments is conducted to evaluate the effects of the two parameterizations

(A15 and SPRAY-GQ) of sea spray-mediated heat flux on ocean, atmosphere and waves in two 56-day
periods, from January 3 to February 28, 2017 and from August 3 to September 28, 2018 for boreal winter
and boreal summer, respectively. For each period, two sensitivity experiments are carried out. The first
is the SPRAY-A15 experiment, in which A15 is used with two-way fully coupling. The second is the
SPRAY-GQ experiment, in which SPRAY-GQ parameterization is used instead of A15.
**4 Results**
**4.1 Comparison with A92**

Based on the daily global WSP10, 2-m air temperature, 2-m dewpoint temperature, surface pressure

and SWH of ERA5, the daily global OISST, and the ESA monthly global salinity, $H_{S,SP}$ and $H_{L,SP}$
from A15, SPRAY-GQ and A92 are calculated (Fig. 3). The computational time for SPRAY-GQ is about
the same as that for A15, and about 36 times less than the time for A92. Compared with A92 (the black
dotted line), A15 (red) overestimates $H_{S,SP}$ for low $H_{S,SP}$ (<50 W/m$^2$) and underestimates $H_{S,SP}$ for
high $H_{S,SP}$ (>50 W/m$^2$) with a root mean square error (RMSE) of 3.40 W/m$^2$ (Fig. 3a), while A15 shows
consistent overestimations with a RMSE of 2.98 W/m$^2$ for $H_{L,SP}$ (Fig. 3b). Overall, the RMSE of A15
is about 2.69 W/m$^2$ for sea-spray mediated total heat flux ($TH_{SP} = H_{S,SP}+H_{L,SP}$; Fig. 3c). Compared with
A15, SPRAY-GQ (blue) has less deviation from A92 for both $H_{S,SP}$ and $H_{L,SP}$ (Fig. 3a&b). The
corresponding RMSEs of SPRAY-GQ for $H_{S,SP}$, $H_{L,SP}$ and $TH_{SP}$ are 0.83 W/m$^2$, 0.92 W/m$^2$ and 0.62
W/m$^2$, all significantly lower (P<0.05 in Student's t-test) than those of A15.

To test robustness of the results, we also use WSP10, 2-m air temperature and 2-m specific humidity

of OAFlux dataset to estimate $H_{S,SP}$ and $H_{L,SP}$. As shown in Fig. 4, SPRAY-GQ has significantly
(P<0.05 in Student's t-test) lower deviations and RMSEs than A15, consistent with Fig. 3. Note that the
values of $H_{S,SP}$ and $H_{L,SP}$ in Fig.4 are larger than those in Fig. 3, since the equivalent neutral wind





speed from OAFlux is generally overestimated compared to the observed wind speed (Seethala et al.,
2021; Praveen Kumar et al., 2012). In addition, since it is common to derive SWH from empirical
equations (e.g., Andreas et al., 2008; Andreas et al., 2015; Andreas and Decosmo, 2002; Andreas, 1992),
we also use SWH generated by empirical equations of WSP10 (Andreas, 1992) instead of ERA5 SWH
to estimate $H_{S,SP}$ and $H_{L,SP}$ (Fig. 5). Again, the RMSEs decrease significantly (P<0.05 in Student's t-
test) in SPRAY-GQ compared to A15, though the RMSEs become higher for all estimates due to the
enhanced biases of SWH. Thereby, it is clear that the performance of SPRAY-GQ is always better than
A15. Next, we will evaluate and compare the two fast algorithms in an atmosphere-ocean-wave coupled
system (CFSv2.0-WW3).
**4.2 Comparison in the CFSv2.0-WW3 Coupling System**
In this section, comparisons are made for simulated SSTs, WSP10s as well as SWHs against OISST
and ERA5 reanalysis (Figs. 6-11). The results in the first three days are excluded in the comparison, since
the wave influences are weak at the beginning of the simulations. The computational time is about the
same for experiments SPRAY-GQ and SPRAY-A15.
**4.2.1 Sea Surface Temperature (SST)**
In the austral summer, compared with OISST, large SST biases (>1 ℃ or <-1 ℃) of SPRAY-A15
occur in the Southern Hemisphere (SH; Fig. S1a in supplementary), especially in the Southern Ocean. It
is always a challenge for reducing the large SST biases in the Southern Ocean for climate models (e.g.,
Alessandro et al., 2019; Wang et al., 2014; Li et al., 2013; Bodas-Salcedo et al., 2012; Ceppi et al., 2012).
In Fig. 6a, SSTs north (south) of 50°S in experiment SPRAY-A15 are mainly underestimated
(overestimated). The domain-averaged RMSE (0-360°E, 40-75°S) increases in the first month and then
levels off (red line in Fig. 6c). While the domain-averaged RMSE in experiment SPRAY-GQ levels off
about a week earlier (black line in Fig. 6c). The time series of RMSE in SPRAY-GQ is significantly
lower than that in SPRAY-A15 (P<0.05 in Student's t-test). The decreased SST RMSE in SPRAY-GQ
is resulted from the increased (decreased) SSTs north (south) of 50°S (Fig. 6b).
To understand the effects of sea spray droplets on SST, we calculate the total heat flux (TH=$H_{S,T}$+$H_{L,T}$)





differences between SPRAY-GQ and SPRAY-A15 (Fig. 12g). The TH differences are significantly
correlated with SST differences (Fig. S1b in the supplementary), with the spatial correlation coefficient
of -0.41 (P<0.05 in Student's t-test). We further decompose direct and indirect effects of sea spray
droplets on heat fluxes following Song et al. (2022). The direct effect ($H_{S,SP}$ and $H_{L,SP}$) is induced
directly by sea spray droplets, calculated from A15 (Eqn. B1-B4 of Appendix B) and SPRAY-GQ
(Section 3.1). The indirect effect ($H_S$ and $H_L$) is the heat flux variation induced by changes of
atmosphere and ocean variables (including wind, pressure, humidity and temperature) caused by direct
effect, estimated by subtracting $H_{S,SP}$ and $H_{L,SP}$ from the output heat fluxes ($H_{S,T}$ and $H_{L,T}$) of
experiment SPRAY-A15 and SPRAY-GQ.
In the Southern Ocean, although direct differences of $H_{S,SP}$ and $H_{L,SP}$ are relatively small (<10
W/m², Fig. 12b, e, &h), the resulting changes of temperature and humidity lead to relatively large
differences in indirect effects of $H_S$ and $H_L$ (Fig. 12c, f, &i). Enhanced (reduced) $TH_{S,SP}$ from ocean
to atmosphere in the summer leads to increased (decreased) air-sea temperature difference and thus
enhances (weakens) $H_S$. Meanwhile the warmer (cooler) air also causes more (less) evaporation and thus
more (less) $H_L$. Finally, the enhanced (reduced) TH cools (warms) SST.
In the boreal summer, large SST biases (>1 ℃ or <-1 ℃) of SPRAY-A15 mainly occur at mid-high
latitudes of the Northern Hemisphere (NH; Fig. S2a in supplementary). Significant underestimations
occur in the western and northern part of the North Pacific and at mid latitudes of the North Atlantic,
while large positive SST biases mainly occur in the eastern part of the North Pacific and at high latitudes
of the North Atlantic (Fig. 7a). In experiment SPRAY-GQ, SSTs are warmer (cooler) in the previously
underestimated (overestimated) regions (Fig. 7b). Therefore, the domain-averaged RMSE (0-360°E, 20-
75°N) in SPRAY-GQ is significantly lower (P<0.01 in Student's t-test) than in SPRAY-A15 after the
first three weeks (Fig. 7c). The spatial correlation coefficient between TH differences and SST
differences (Fig. 13g&Fig. S2b) is -0.32 (P<0.05 in Student's t-test). Consistent with the austral summer,
the SST changes are related to the changes of heat flux (Fig. 13). The indirect effects of latent heat flux
(Fig. 13c) play a major role in TH differences, which are modified by the direct effects (Fig. 13b, e, &h).
In addition, the changes of surface wind also contribute to the changes of SST. The enhanced (reduced)
winds lead to stronger (weaker) ocean mixing, and thus cooler (warmer) SST (Fig. S3&S4).





**4.2.2 10-m Wind Speed (WSP10) and Significant Wave Height (SWH)**
Compared with experiment SPRAY-A15, significant improvements of WSP10 in SPRAY-GQ occur
at mid-low latitudes of the NH (0-360°E, 0-60°N) in both winter and summer (Fig.8&9). The domain-
averaged bias of WSP10 (SPRAY-A15 minus ERA5) is 0.37 m/s and 0.24 m/s in winter and summer,
respectively, mainly due to the overestimations over the Pacific and the Atlantic Ocean (red in
Fig.8a&9a). Whereas in SPRAY-GQ, the domain-averaged bias (SPRAY-GQ minus ERA5) is 0.26 m/s
and 0.03 m/s in winter and summer respectively. The domain-averaged RMSEs of WSP10s increase with
time in the first two weeks and then gradually level off (Fig. 8c&9c). The differences of WSP10 RMSEs
between SPRAY-GQ (black) and SPRAY-A15 (red) are very small in the first two weeks. Afterwards
the time series of RMSE in SPRAY-GQ is lower than that in SPRAY-A15 significantly at 99%
confidence level in both boreal winter (Fig. 8c) and boreal summer (Fig. 9c).
The simulated SWHs changes are closely related to the changes of WSP10 (Shi et al. 2022). Therefore,
the differences of SWHs (Fig.10&11) are consistent with those of WSP10s (Fig.8&9), with
overestimated (underestimated) WSP10s corresponding to overestimated (underestimated) SWHs
compared with ERA5. The SWHs in SPRAY-GQ improve compared with those in SPRAY-A15 (Fig.
10b&11b). In winter (summer), the SWH RMSE averages for SPRAY-A15 and SPRAY-GQ are 1.31 m
(0.98 m) and 1.23 m (0.87 m), and after the first two weeks the time series of RMSE in SPRAY-GQ is
lower than that in SPRAY-A15 significantly at 99% confidence level in both winter (Fig. 10c) and
summer (Fig. 11c).
The direct and indirect effects of sea spray droplets on heat fluxes can influence estimates of WSP10
and then SWH. The changes of WSP10s are related to the direct effects ($H_{S,SP}$ and $H_{L,SP}$; Fig. 12b, e,
&h; Fig. 13b, e, &h). The spatial correlation coefficients between WSP10 differences (Fig. S3b&S4b)
and $TH_{SP}$ differences (Fig. 12h&13h) are 0.51 and 0.69 (P<0.01 in Student's t-test) in winter and
summer, respectively. Because the directly increased (decreased) heat fluxes enhance (reduce)
turbulence, promote (hinder) the downward transmission of momentum from the upper layer of
atmosphere, and then accelerate (decelerate) the surface wind speed (Wallace et al., 1989). While the
accelerated (decelerated) WSP10s further result in increased (decreased) interfacial heat transport ($H_S$,
$H_L$), as well as increased (decreased) SWHs.





**5 Conclusions and Discussions**
Based on a GQ method, we develop a new fast algorithm based on Andreas's (1989, 1990, 1992) full-
size microphysical parameterization (A92) for sea spray-mediated heat fluxes. Using global satellite
measurements and reanalysis data, SPRAY-GQ parameterization is validated to approximate A92 more
accurately than the A15 fast algorithm (Andreas et al., 2015). To evaluate the SPRAY-GQ/A15
parameterization, we implement them in the two-way coupled CFSv2.0-WW3 system. A series of 56-
day simulations from January 3 to February 28, 2017 and from August 3 to September 28, 2018 are
conducted. The results are compared against OISST satellite measurements and ERA5 reanalysis. The
comparison shows that the sea spray-mediated heat flux in SPRAY-GQ can reasonably modulate total
heat flux, significantly improve the SST biases in the Southern Ocean (mid-high latitudes of the NH) for
the austral (boreal) summer, as well as WSP10 and SWH at mid-low latitudes of the NH for both boreal
winter and summer. Overall, our fast algorithm based on GQ is applicable to sea spray-mediated heat
flux parameterization in coupled models.
In addition to the variables aforementioned, the changes of simulated cloud fraction were also
compared. However, the effects of sea spray-mediated heat flux on cloud fraction are non-significant for
the 2-month simulation, so the results are not shown. Besides, for simulated WSP10 and SWH, the
SPRAY-GQ parameterization used in the study mainly improves the biases at mid-low latitudes of the
NH, while the significant overestimations in the SH are only slightly improved (Fig. S3-S6 in
supplementary). As Andreas (2004) indicated, sea spray droplets also influence the surface momentum
flux by injecting more momentum into the ocean from the atmosphere, which might further decrease the
surface wind speed. We will consider this process in the future study.
Sea spray-mediated heat fluxes are sensitive to the sea spray generation function $dF/dr_0$. Based on
a number of laboratory and field observations, varieties of $dF/dr_0$ were derived (e.g., Koga, 1981;
Monahan et al., 1982; Troitskaya et al., 2018; Andreas, 1992, 1998, 2002; Fairall et al., 1994; Veron,
2015), whereas their differences can reach six orders of magnitude (Andreas, 1998). There is currently
no consensus on the most suitable choice. In this study, we use $dF/dr_0$ of Fairall et al. (1994),
recommended by Andreas (2002). It is also consistent with recent observations of Xu et al. (2021b).
Since the new scheme based on GQ is independent of sea spray generation function, the new scheme can





also be applied to sea spray-mediated heat fluxes estimation with different $dF/dr_0$.
**Appendix A**
**Microphysical Parameterization of A92**
Based on the cloud microphysical parameterization of Pruppacher and Klett (1978), Andreas (1989,
1990, 1992) proposed a parameterization of sea spray-related heat fluxes for droplets with different radius,
from formation at sea surface to equilibrium with environment, that is,

$$Q_S = \rho_w C_{ps}(T_w - T_{eq})\left[1 - \exp\left(-\frac{\tau_f}{\tau_T}\right)\right]\left(\frac{4\pi r_0^3}{3}\frac{dF}{dr_0}\right), \quad (A1)$$

$$Q_L = \begin{cases} \rho_w L_v\left\{1 - [\frac{r(\tau_f)}{r_0}]^3\right\}\left(\frac{4\pi r_0^3}{3}\frac{dF}{dr_0}\right), \tau_f \leq \tau_r, \\ \rho_w L_v\left\{1 - (\frac{r_{eq}}{r_0})^3\right\}\left(\frac{4\pi r_0^3}{3}\frac{dF}{dr_0}\right), \tau_f > \tau_r. \end{cases} \quad (A2)$$

Here $Q_S$, $Q_L$ are sensible heat flux and latent heat flux resulted by sea spray droplets with initial radius
$r_0$, $\rho_w$ is the sea water density, $C_{ps}$ is the specific heat, $L_v$ is the latent heat of vaporization of water,
$T_w$ is the water temperature, $T_{eq}$ is the temperature of droplet when it reaches thermal equilibrium with
ambient condition, $r_{eq}$ is the radius of droplet when it reaches moisture equilibrium with ambient
condition, $\tau_f$ is the residence time for droplets in the atmospheric, $r(\tau_f)$ is the corresponding radius,
$\tau_T$ is the characteristic e-folding time of droplet temperature, and $\tau_r$ is the characteristic e-folding time
of droplet radius. The detailed calculation of these microphysical quantities can be found in Andreas
(1989, 1990, 1992). And $dF/dr_0$ is the sea spray generation function, which represents the number
produced of droplets with initial radius $r_0$ (Andreas, 1992). For this term, the function of Fairall et al.
(1994) was recommended by Andreas (2002). According to the review in Andreas (2002), the $dF/dr_0$
of Fairall et al. (1994) is related on that of Andreas (1992) as

$$\frac{dF}{dr_0} = 38 \times 3.84 \times 10^{-6} U_{10}^{3.41} r_0^{-0.024} \frac{dF_{A92}}{dr_{80}}\bigg|_{U_{10}=11\ m/s}, \quad (A3)$$

$$\frac{dF_{A92}}{dr_{80}}\bigg|_{U_{10}=11\ m/s} =$$

$$\begin{cases} e^{(4.405 - 2.646(log r_{80}) - 3.156(log r_{80})^2 + 8.902(log r_{80})^3 - 4.482(log r_{80})^4)}, r_{80} \leq 15\mu m; \\ 1.02 \times 10^4 r_{80}^{-1}, 15 \leq r_{80} \leq 37.5\mu m; \\ 6.95 \times 10^6 r_{80}^{-2.8}, 37.5 \leq r_{80} \leq 100\mu m; \\ 1.75 \times 10^{17} r_{80}^{-8}, r_{80} \geq 100\mu m \end{cases} \quad (A4)$$

Here $U_{10}$ is the 10-m wind, $r_{80} = 0.518 r_0^{0.976}$.



The total sea spray fluxes are obtained by integrating $Q_S$ and $Q_L$ corresponding to all $r_0$. Based on
Andreas (1990), the lower and upper limits of $r_0$ is $2\mu m$ and $500\mu m$, that is,

$$\overline{Q_S} = \int_2^{500} Q_S(r_0)dr, \qquad (A5)$$

$$\overline{Q_L} = \int_2^{500} Q_L(r_0)dr. \qquad (A6)$$

While $\overline{Q_S}$ and $\overline{Q_L}$ are nominal sea spray fluxes but not the actual $H_{S,SP}$ and $H_{L,SP}$ (Andreas and
Decosmo, 1999, 2002), because there are interactions between these two terms and the microphysical
functions also lead to uncertainties (Fairall et al., 1994). Therefore, $\overline{Q_S}$ and $\overline{Q_L}$ are tuned by non-
negative constants α, β and γ (Andreas and Decosmo, 2002; Andreas et al., 2008; Andreas et al., 2015;
Andreas, 2003) as

$$H_{S,SP} = \beta\overline{Q_S} - (\alpha - \gamma)\overline{Q_L}, \qquad (A7)$$

$$H_{L,SP} = \alpha\overline{Q_L}. \qquad (A8)$$

In Eqn. (A8), the α term indicates the sea spray-mediated latent heat flux from the top of DEL to
atmosphere. Because the evaporation of droplets absorbs heat, which is provided by sea spray-mediated
sensible heat (Fairall et al., 1994), the negative α term appears in Eqn. (A7). Whereas the evaporation
also cools DEL and thus increases the air-sea temperature difference, therefore it contributes to a positive
γ term in Eqn. (A7). Different values of α, β and γ were given in Andreas and Decosmo (2002),
Andreas (2003), Andreas et al. (2008) and Andreas et al. (2015), to minimize the bias between
estimations and observations of turbulent heat fluxes measured by eddy correlation. And Andreas et al.
(2015) validated the most observation data, which are 4000 sets, to derive $\alpha = 2.46, \beta = 15.15, \gamma = $
$1.77$.
**Appendix B**
**Fast Algorithm of A15**
Andreas (2003) and Andreas et al. (2008, 2015) developed a fast algorithm to approximate $H_{S,SP}$,
$H_{L,SP}$ by a characteristic radius, that is,

$$H_{S,SP} = \beta\overline{Q_S} - (\alpha - \gamma)\overline{Q_L} \approx \rho_w C_{ps}(T_W - T_{eq,100})V_s(u_*), \qquad (B1)$$

$$H_{L,SP} = \alpha\overline{Q_L} \approx \rho_w L_v \left\{ 1 - [\frac{r(\tau_{f,50})}{50\mu m}]^3 \right\} V_L(u_*). \qquad (B2)$$



Here $T_{eq,100}$ is $T_{eq}$ of droplets with $r_0$=100 μm, $\tau_{f,50}$ is $\tau_f$ of droplets with $r_0$=50 μm, $V_s$ and
$V_L$ are functions of the bulk friction velocity $u_*$. As indicated by Andreas et al. (2008, 2015), the
characteristic radiuses of 100 μm and 50 μm for sensible and latent heat fluxes are chosen,
respectively, because $Q_S$ and $Q_L$ show a large peak in the vicinity of these values (Fig. 1). $V_s$ and $V_L$
are calculated in Andreas et al. (2015) as

$$V_S = \begin{cases} 3.92 \times 10^{-8}, & 0 \leq u_* \leq 0.1480 \; m/s \\ 5.02 \times 10^{-6} u_*^{2.54}, & u_* \geq 0.1480 \; m/s \end{cases}, \tag{B3}$$

$$V_L = \begin{cases} 1.76 \times 10^{-9}, & 0 \leq u_* \leq 0.1358 \; m/s \\ 2.08 \times 10^{-7} u_*^{2.39}, & u_* \geq 0.1358 \; m/s \end{cases}. \tag{B4}$$

**Appendix C**
**Gaussian Quadrature (GQ)**
GQ is a method to approximate the definite integral of a function $f(x)$ via the function values at a
small number of specified nodes (Gauss, 1815; Jacobi, 1826). In this study we use the form of n-node
Gauss–Legendre quadrature on [-1, 1] as

$$\int_{-1}^{1} f(x)dx \approx \sum_{i=1}^{n} \omega_i f(x_i). \tag{C1}$$

Here $x_i$ is the specified node, and $\omega_i$ is the corresponding weight. For n=3, $x_1$=-0.775, $x_2$=0,
$x_3$=0.775, $\omega_1$=$\omega_3$=0.556, $\omega_2$=0.889.
While for a function $g(\xi)$ on [a, b], Eqn. (C1) can be transformed to

$$\int_{a}^{b} g(\xi)d\xi = \int_{-1}^{1} g\left(\frac{b-a}{2}x + \frac{a+b}{2}\right)\frac{d\xi}{dx}dx$$
$$\approx \frac{b-a}{2}\sum_{i=1}^{n} \omega_i g\left(\frac{b-a}{2}x_i + \frac{a+b}{2}\right). \tag{C2}$$

**Code and data availability**
The code of sea spray can be found under https://doi.org/10.5281/zenodo.7100345 (Shi and Xu, 2022).
The code for CFSv2.0-WW3 system can be found under https://doi.org/10.5281/zenodo.5811002 (Shi et
al., 2021) including the coupling, preprocessing, run control and postprocessing scripts. The initial fields
for CFSv2.0 are generated by the real time operational Climate Data Assimilation System, downloaded
from the CFSv2.0 official website (http://nomads.ncep.noaa.gov/pub/data/nccf/com/cfs/prod). The daily



average satellite Optimum Interpolation SST (OISST) data are obtained from NOAA
(https://www.ncdc.noaa.gov/oisst). The fifth generation European Centre for Medium-Range Weather
Forecasts (ECMWF) Reanalysis (ERA5) are available at the Copernicus Climate Change Service (C3S)
Climate Date Store (https://cds.climate.copernicus.eu/cdsapp#!/dataset/reanalysis-era5-single-levels).
The daily Objectively Analyzed air-sea Fluxes (OAFlux) products are available at
https://oaflux.whoi.edu/heat-flux. The global monthly mean salinity observation of European Space
Agency (ESA) are from https://climate.esa.int.

**Author contribution**

FX and RS designed the experiments and RS carried them out. RS developed the code of coupling
parametrizations and produced the figures. RS prepared the manuscript with contributions from all co-
authors. FX contributed to review and editing.

**Acknowledgments**

This work was supported by the National Key Research and Development Program of China
(2020YFA0607900, 2021YFC3101601), and the National Natural Science Foundation of China
(42176019). We also thank Dr. Jiangnan Li for help of GQ codes.

**Competing Interests**

The contact author has declared that neither they nor their co-authors have any competing interests.

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



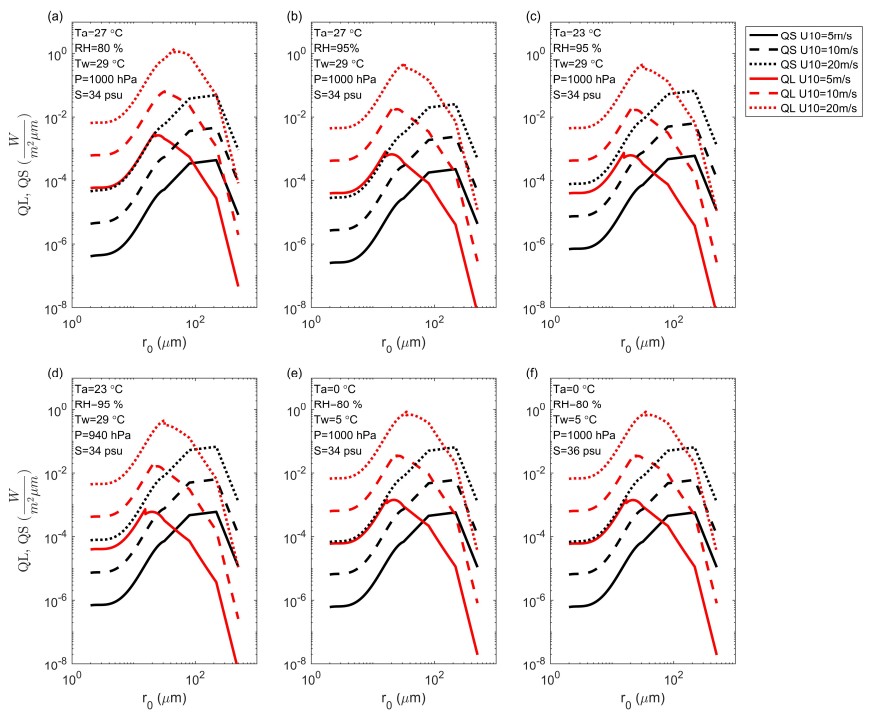

**Figure 1**. The radius-specific sea spray-mediated sensible ($Q_S$; black) and latent ($Q_L$; red) heat fluxes as functions of initial radius $r_0$: U10, Ta, RH, Tw, P and S are 10-m wind speed, 2-m air temperature, 2-m relative humidity, sea surface temperature, surface air pressure and surface salinity, respectively.



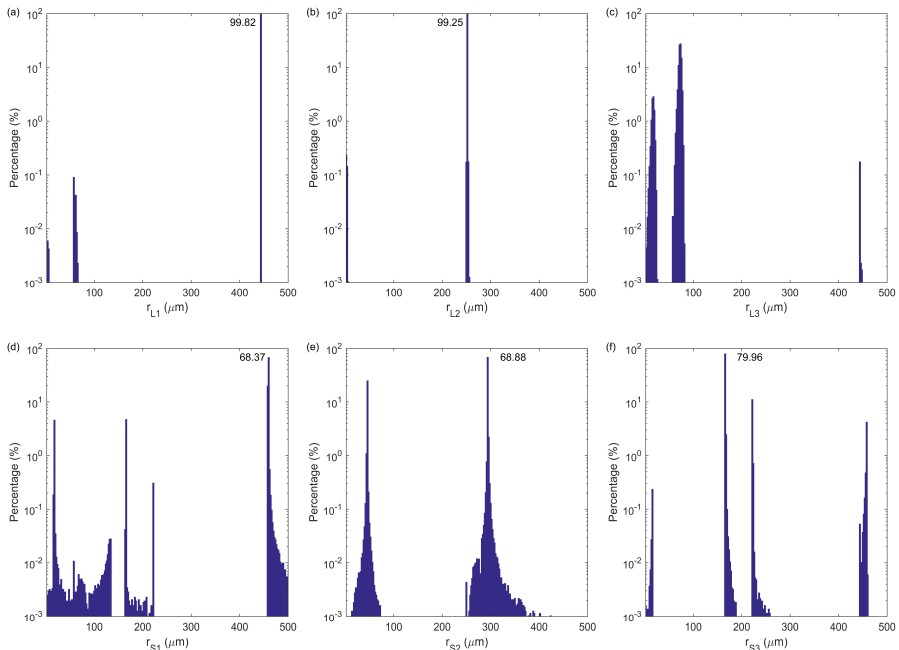

**Figure 2**. The distribution of occurrence frequency in percentage for GQ radius nodes: (a) the first node of latent heat flux; (b) the second node of latent heat flux; (c) the third node of latent heat flux; (d) the first node of sensible heat flux; (e) the second node of sensible heat flux; (f) the third node of sensible heat flux. The peak frequencies are marked.



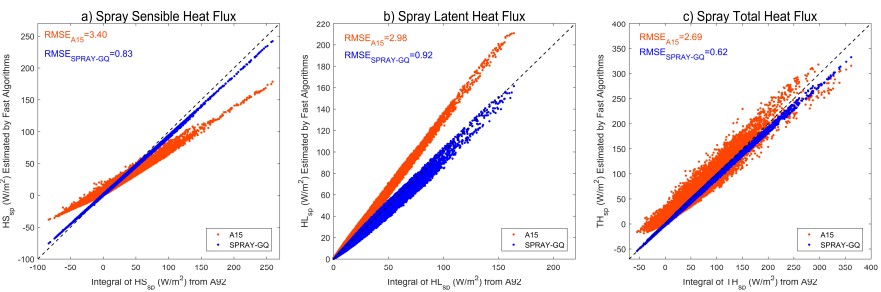

**Figure 3**. Scatter plots of $H_{S,SP}$ (a), $H_{L,SP}$ (b) and total heat flux $TH_{SP} = H_{S,SP}+H_{L,SP}$ (c) estimated by fast algorithms (y-axis) vs those estimated by spectral integral in microphysical parameterization (x-axis): The dotted black line is y=x. The corresponding RMSEs are marked in the upper left corner.

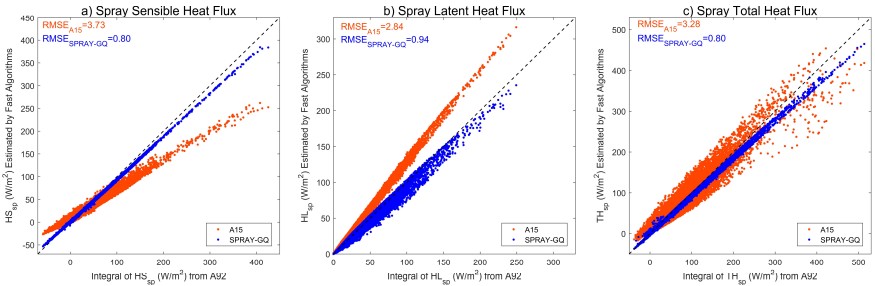

**Figure 4**. The same as Figure 3, but WSP10, 2-m air temperature and 2-m specific humidity of OAFlux are used.

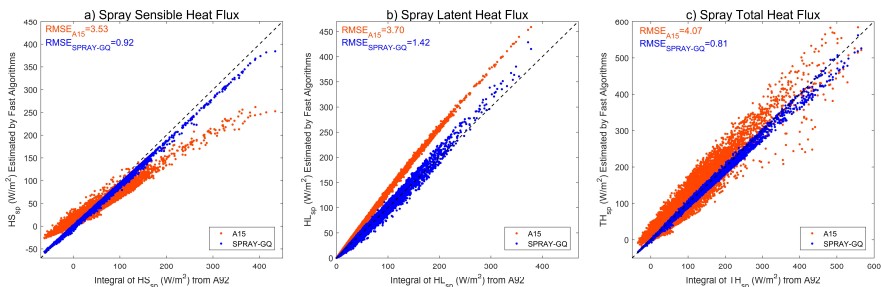

**Figure 5**. The same as Figure 4, but SWH are derived by WSP10 instead of ERA5 SWH.



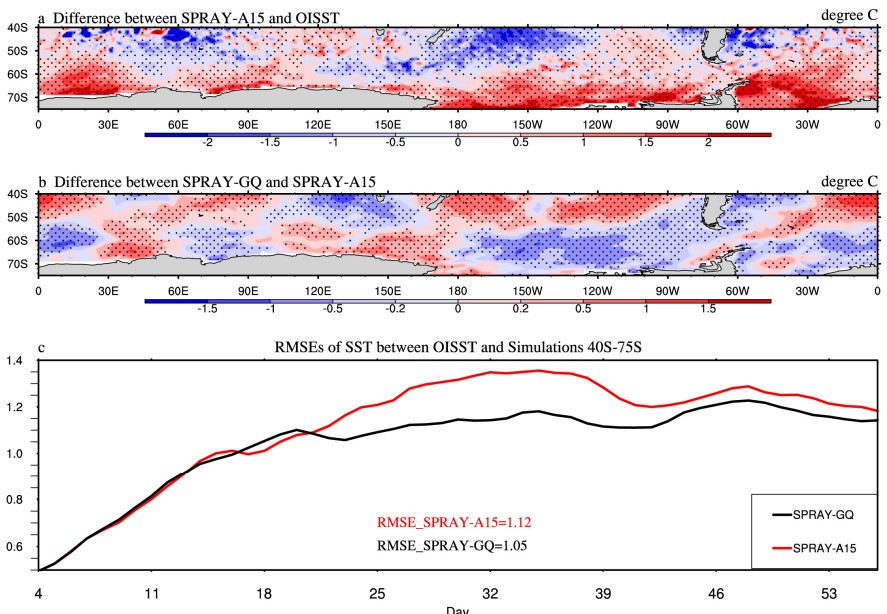

**Figure 6**. The 53-day average SST (℃) differences between SPRAY-A15 and OISST (a; SPRAY-A15 minus OISST), the differences between SPRAY-GQ and SPRAY-A15 (b; SPRAY-GQ minus SPRAY-A15), and the time series of domain-averaged RMSE (c; 0-360°E, 40-75°S) in Jan-Feb, 2017. The first 3-day simulation is discarded. The dotted areas are statistically significant at 95% confidence level.

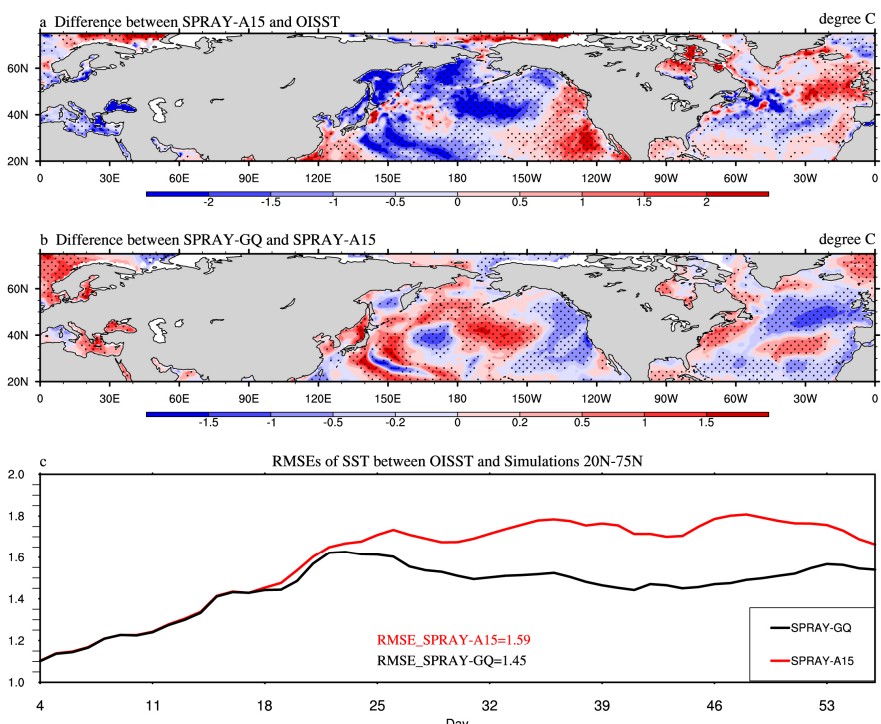

**Figure 7**. The same as Figure 6, but for Aug-Sep, 2018 in 0-360°E, 20-75°N.



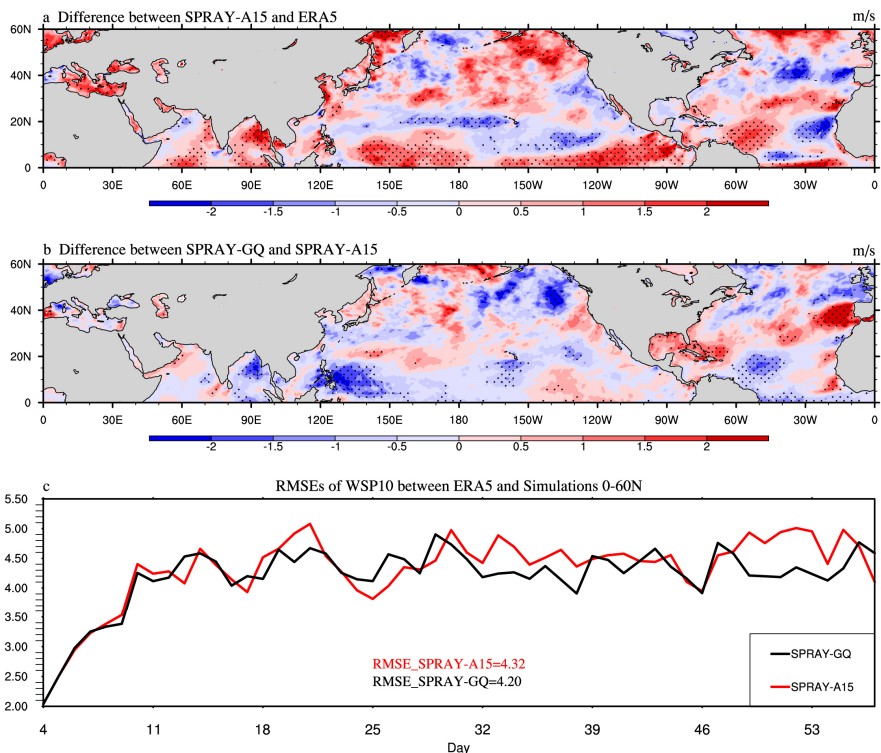

**Figure 8**. The 53-day average WSP10 (m/s) differences between SPRAY-A15 and ERA5 (a; SPRAY-A15 minus ERA5), the differences between SPRAY-GQ and SPRAY-A15 (b; SPRAY-GQ minus SPRAY-A15), and the time series of domain-averaged RMSE (c; 0-360°E, 0-60°N) in Jan-Feb, 2017. The first 3-day simulation is discarded. The dotted areas are statistically significant at 95% confidence level.



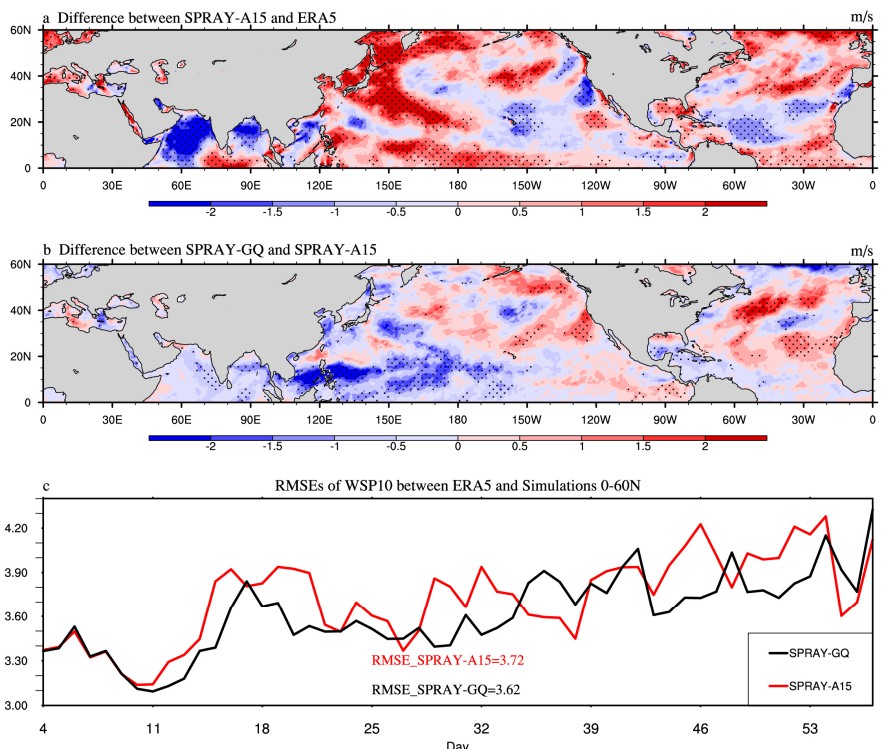

**Figure 9**. The same as Figure 8, but for Aug-Sep, 2018.

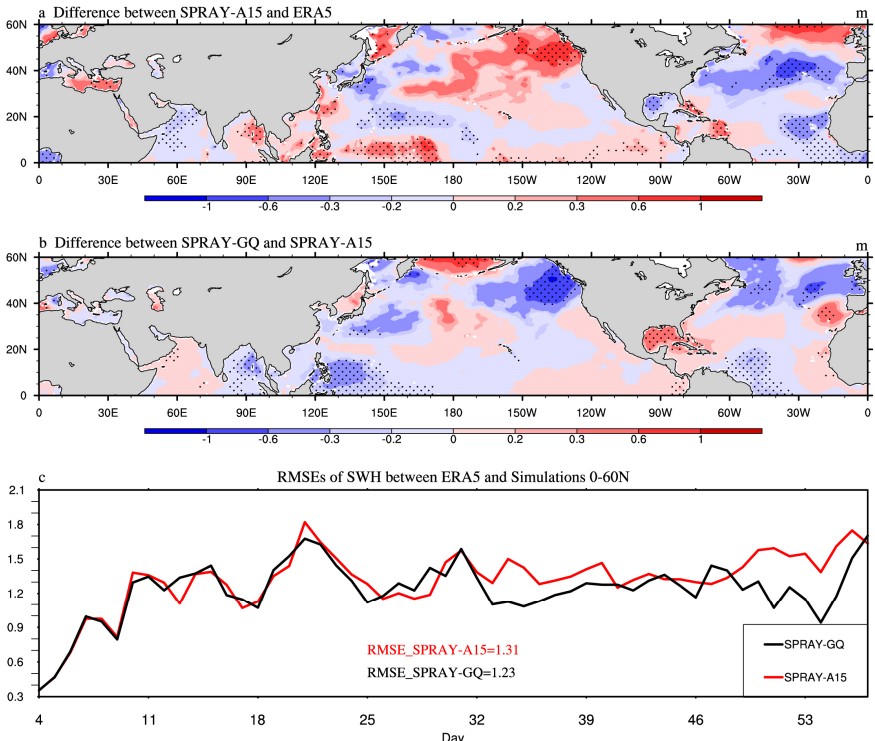

**Figure 10**. The 53-day average SWH (m) differences between SPRAY-A15 and ERA5 (a; SPRAY-A15 minus ERA5), the differences between SPRAY-GQ and SPRAY-A15 (b; SPRAY-GQ minus SPRAY-A15), and the time series of domain-averaged RMSE (c; 0-360°E, 0-60°N) in Jan-Feb, 2017. The first 3-day simulation is discarded. The dotted areas are statistically significant at 95% confidence level.





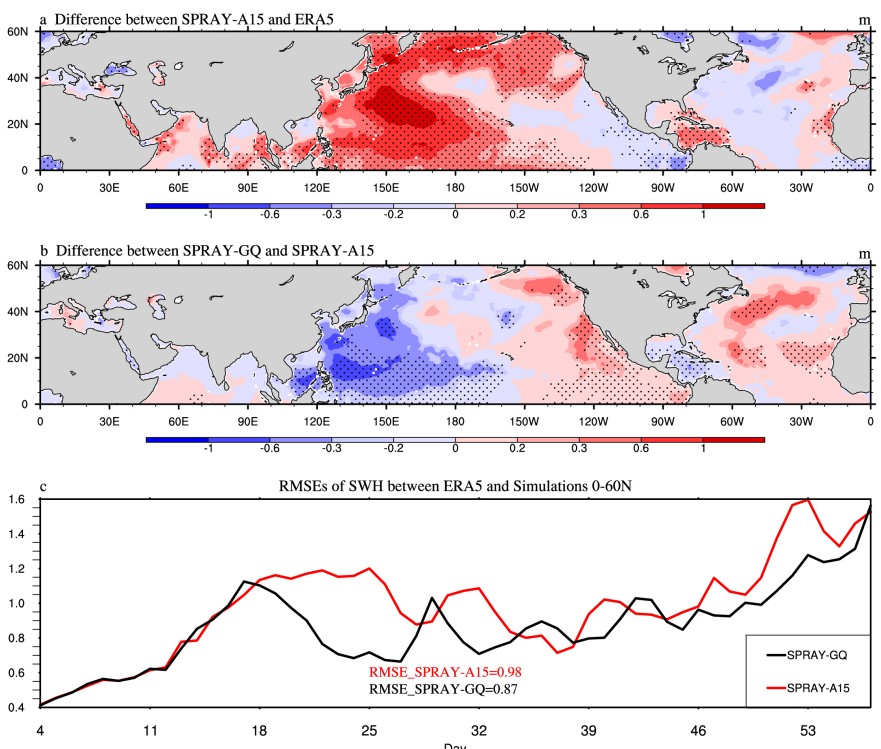

**Figure 11**. The same as Figure 10, but for Aug-Sep, 2018.





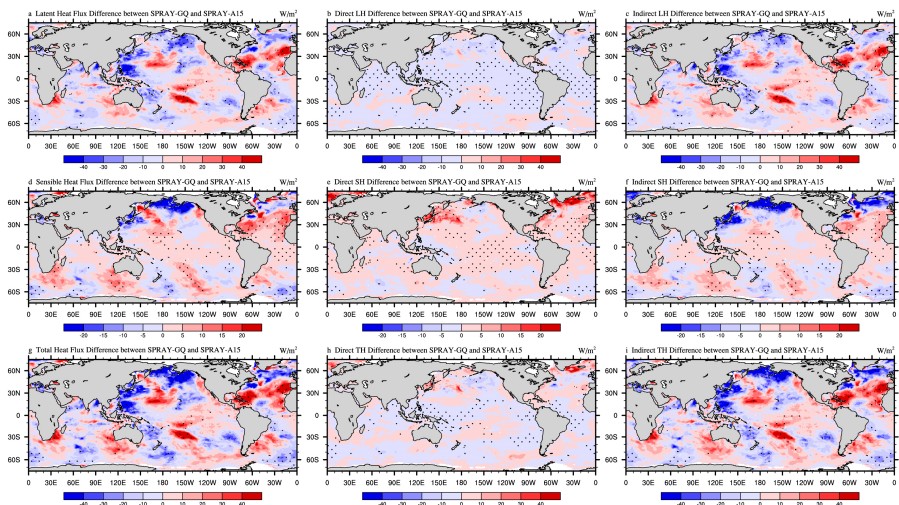

**Figure 12**. The 53-day average differences of latent heat flux (a-c), sensible heat flux (d-f) and total heat flux (g-i) between SPRAY-A15 and SPRAY-GQ (SPRAY-GQ minus SPRAY-A15) in Jan-Feb, 2017. The direct differences indicate sea spray-mediated heat flux differences (b, e, h), and the indirect differences indicate interfacial (bulk) heat flux differences resulted by sea spray (c, f, i). The dotted areas are statistically significant at 95% confidence level. A positive value of flux indicates an upward direction.





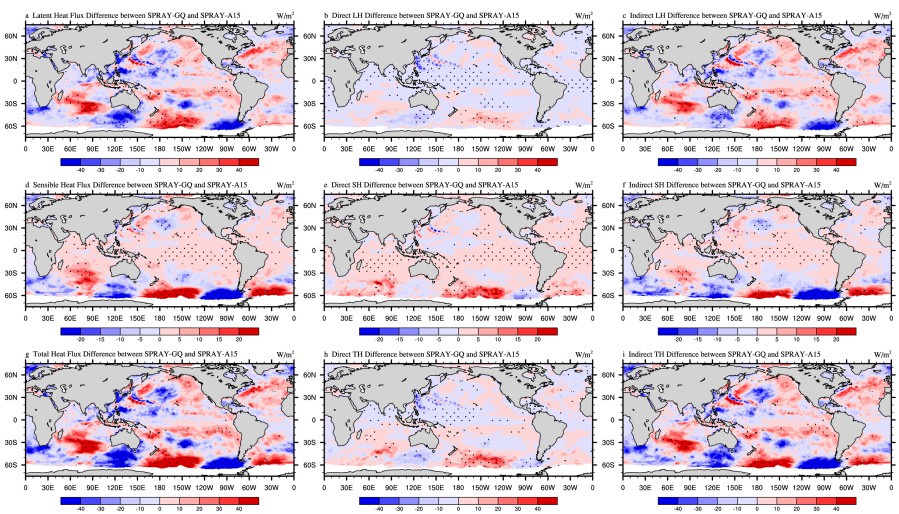

**Figure 13**. The same as Figure 12, but for Aug-Sep, 2018.