# Peer review of "Accelerated Estimation of Sea Spray-Mediated Heat Flux"

_Geoscientific Model Development, 2022_

## Referee Comment (RC1)

**General comments**

In this paper, the Gaussian Quadrature (GQ) method is used to calculate the spray-mediated heat flux, instead of the current full-size spectrum integral (A92) and the fast algorithm (A15). A global atmosphere-ocean-wave coupled model CFSv2.0-WWW3 is employed, and two time periods of 56 days in boreal summer and winter are conducted to test the sensitivies of SST, 10 m wind speed and significant wave height to the new scheme that the authors proposed. Although the improvement on spray-mediated heat flux is not physical, the computational time is about 36 times less than that of A92. In addition, the introduction of this new method improves the simulation of SST, 10 m wind speed (WSPD10) and significant wave height (SWH). Based on above reasons, I think this manuscript can be considered for publication if the authors address all my queries and comments below.

**Specific comments**

1. My first comment is about the title of this manuscript. I think the 'improved' is not appropriate. As I mentioned above, the new method that the author proposed is not physical improvement on spray-mediated heat flux, and it has not been validated against the directly or indirectly observed sea spray heat flux. Although SST, WPSD10 and SWH are has been improved, it can't be said that the sea spay-mediated flux is 'improved'. Perhaps '**Accelerated** Estimation of

Sea Spray-Mediated Heat Flux Using Gaussian Quadrature method and…….' is more in line with the content of this manuscript.

2. Line 57, I do understand the positive effect of sea spray on tropical cyclone, but what is the 'negative effect of enhanced surface drag'? The author needs to be more specific in the text.

3. What are the prerequisites for the use of Gaussian-Legendre quadrature for f(x)? just smooth?

4. Line 118, the authors said 'the sorting leads to high complexity of GQ comparable to A92', thereby the authors try to avoid sorting. However, the authors still sort $Q_S$ and $Q_L$ from largest to smallest (lines 124-125). Is it contradictory?

5. Equations (3) to (5) really make me puzzled. According to the author's description in Appendix C, $r_i = \dfrac{b-a}{2}x_i + \dfrac{a+b}{2}$ , if $x_1 = -0.775, x_2 = 0, x_3 = 0.775$ , and the lower and upper limits of sea spray radius are $a = 2, b = 500$ , I calculate the values of GQ 3-nodes and get the following results $r_1 = 443, r_2 = 251, r_3 = 58$ , which are almost consistent with the 3-nodes values of $Q_L$ given by the authors. How did the authors get the GQ 3-nodes for $Q_S$? why $Q_S$ and $Q_L$ use different GQ 3-nodes? Given that the potential users may be interested in this new method, the authors need to clarify how the 3-nodes $r_{s1}(r_{l1}), r_{s2}(r_{l2}), r_{s3}(r_{l3})$ are obtained by sorting $Q_S$ and $Q_L$ as much detailed as possible.

6. Lines 127-128, How can we see that $r_{l3}$ is related to WPSD10 from

Figure 2c?

7. Figure 2 is very difficult for me to understand. As far as I can understand from the text, if 3 GQ nodes are determined, then 3 percentage values are determined. But in Figure 2, the authors use $r_{s1}(r_{l1}), r_{s2}(r_{l2}), r_{s3}(r_{l3})$ as the name of the x-axis, and there are so many bars in each panel of Figure 2, what do the bars represent? I strongly recommend that the authors devote more space to the introduce the new methods in their manuscript to respond to my comments 5-7.

8. The authors need to briefly describe how the atmospheric and oceanic components of CFSv2.0-WW3 are initialized. I also note that WW3 is not global, then what dataset is used to provide open boundary conditions for WW3?

9. The difference between the spray-mediated heat fluxes calculated by A15 and A92 schemes is so significant. Can the authors comment on what cause this large difference? Extrapolation of $V_S$ and $V_L$ at high wind speeds? or due to the use of single-radius droplets to represent the full-spectrum integral? As Andreas et al (2015) said, $V_S$ & $V_L$ in A15 are extrapolated at high wind speeds, while SSGF (sea spray generation function) in A92 are deduced. The lack of discussion on the causes of this discrepancy will greatly diminish the importance of this article. I am seriously concerned about this.

10. It would be better to superimpose the Mean Error onto Figure

6c,7c,8c,9c,10c, and 11c.

11. According to Andreas et al. (2015), the effect of sea spray become significant at wind speed of 10-13 $ms^{-1}$. How strong can the simulated WSPD10 be? Therefore, I would like to see the global distribution of WSPD10 simulated by CFSv2.0-WW3.

12. Line 178, what does equivalent neutral wind mean? Do the authors mean that the equivalent neutral winds in OAFLux are larger than those in ERA5?

13. Lines 230-231, The expression is not accurate, the reduced wind and weaker mixing can lead to warmer SST?

14. As we know, satellite scatterometer and altimeter data are usually used to validate WSPD10 and SWH for short term weather forecast. I don't know why the authors use ERA5 reanalysis as validation data for seasonal prediction.

15. Does the introduction of sea spray can improve the simulation of other elements? For example, air temperature and humidity.

16. Line 246, there is a grammatical mistake in this sentence. And Fig10b&11b do not support 'The SWHs in SPRAY-GQ improve compared with those in SPRAY-A15'. Do the authors mean that there is a significant difference between SPRAY-GQ and SPRAY-A15?

17. As far as I know, the sea spray algorithm codes of Andreas are open source, please upload the author's modified A92 codes to a repository

so that others can repeat the author's results.

18.  Lines 255-259, the authors try to discuss the physical mechanism that responsible for the accelerated surface wind. However, the citation does not seem to support the author's conclusions. I can understand that the increase of air-sea heat flux could promote air convection in the vertical, but how does it promote the downward transmission of momentum from the upper layer of atmosphere? By affecting the large-scale atmospheric circulation? Please provide appropriate citations or give your own analysis to support your points.

19. Lines 263-264, again, without verification by direct/indirect observation, we can't say that SPRAY-GQ is more accurate than A15. All we can say is that the difference between SPRAY-GQ and A92 is smaller than that between A15 and A92.

20. Finally, I think the current experimental design is insufficient, a reference experiment without sea spray effect is missing in manuscript. Although it may be expensive to conduct a new set of experiment, it makes sense for the scientific community to understand the importance of spray-mediated heat flux for seasonal and intra-seasonal prediction.

---

## Author Response (AR2)

We sincerely appreciate the reviewer for her/his constructive comments on the manuscript. Our responses are listed as follows in blue. Text is revised accordingly.

**Review from Referee #1**

*In this paper, the Gaussian Quadrature (GQ) method is used to calculate the spray-mediated heat flux, instead of the current full-size spectrum integral (A92) and the fast algorithm (A15). A global atmosphere-ocean-wave coupled model CFSv2.0-WW3 is employed, and two time periods of 56 days in boreal summer and winter are conducted to test the sensitivities of SST, 10 m wind speed and significant wave height to the new scheme that the authors proposed. Although the improvement on spray-mediated heat flux is not physical, the computational time is about 36 times less than that of A92. In addition, the introduction of this new method improves the simulation of SST, 10 m wind speed (WSP10) and significant wave height (SWH). Based on above reasons, I think this manuscript can be considered for publication if the authors address all my queries and comments below.*

*Specific comments*

*1. My first comment is about the title of this manuscript. I think the 'improved' is not appropriate. As I mentioned above, the new method that the author proposed is not physical improvement on spray-mediated heat flux, and it has not been validated against the directly or indirectly observed sea spray heat flux. Although SST, WPSD10 and SWH are has been improved, it can't be said that the sea spay-mediated flux is 'improved'. Perhaps 'Accelerated Estimation of Sea Spray-Mediated Heat Flux Using Gaussian Quadrature method and…….' is more in line with the content of this manuscript.*
Response: We agree. The title is revised as suggested.

*2. Line 57, I do understand the positive effect of sea spray on tropical cyclone, but what is the 'negative effect of enhanced surface drag'? The author needs to be more specific in the text.*
Response: We apologize for the unclear writing. The strong winds and high waves induced by tropical cyclones can enhance sea surface roughness and thus surface drag coefficients. Subsequently, the enhanced surface drag tends to reduce tropical cyclone intensity. In addition, when a sea spray droplet is released into the air, it is accelerated by surface winds, leading to more dissipation of tropical cyclone kinetic energy (Andreas and Emanuel, 2001). So there are negative effects. The text is revised accordingly to clarify in Line 58-63 (Line 61-70 for text with changes tracked).

*3. What are the prerequisites for the use of Gaussian-Legendre quadrature for f(x)? just smooth?*
Response: Yes, just smooth. According to McClarren (2018), the Gaussian-Legendre quadrature can be used to approximate the integral of any f(x) on [a, b] only when f(x) is a smooth function. Text is revised accordingly to clarify in Line 121-122 (Line 135-136 for text with changes tracked).

*4. Line 118, the authors said 'the sorting leads to high complexity of GQ comparable to A92', thereby the authors try to avoid sorting. However, the authors still sort QS and QL from largest to*

*smallest (lines 124-125). Is it contradictory?*

Response: We apologize for the unclear writing. In lines 124-125, we sort $Q_S$ and $Q_L$ to derive Eqn. 3-5 to get the corresponding GQ nodes. Afterwards, we don't sort anymore. The newly-derived GQ nodes can be applied directly. Text is revised to clarify in Line 130-145 (Line 144-164 for text with changes tracked).

*5. Equations (3) to (5) really make me puzzled. According to the author's description in Appendix*

*C, $r_i = \frac{b-a}{2} x_i + \frac{a+b}{2}$, if $x_1$=-0.775, $x_2$=0, $x_3$=0.775, and the lower and upper limits of sea spray*

*radius are a=2, b=500, I calculate the values of GQ 3-nodes and get the following results $r_1$=443,*
*$r_2$=251, $r_3$=58, which are almost consistent with the 3-nodes values of $Q_L$ given by the authors.*
*How did the authors get the GQ 3-nodes for $Q_S$? Why $Q_S$ and $Q_L$ use different GQ 3-nodes? Given*
*that the potential users may be interested in this new method, the authors need to clarify how the 3-*
*nodes $r_{S1}(r_{L1})$, $r_{S2}(r_{L2})$ and $r_{S3}(r_{L3})$ are obtained by sorting $Q_S$ and $Q_L$ as much detailed as*
*possible.*

Response: Yes, according to Appendix C, the three nodes for GQ in this case are $m_1$=443, $m_2$=251, $m_3$=58. Noticeably, the method should be applied only for a smooth function, which is the prerequisite for GQ. So we sort $Q_S(r)$ and $Q_L(r)$ first to obtain the smooth functions, and then calculate the values of $r$ corresponding to their nodes. Since the original $Q_L$ is approximately smooth, the 3-nodes of $Q_L$ in Eqn. 4-5 are almost consistent with the $m_1$, $m_2$ and $m_3$ estimated from Appendix C. On the other hand, the original $Q_S$ is not smooth, as shown in Figure 1. Therefore, after sorting the GQ 3-nodes for $Q_S$ are different with those for $Q_L$.

After sorting local $Q_S(r)$ and $Q_L(r)$ from the largest to the smallest, we obtain $Q_{S\_sort}(m)$ and $Q_{L\_sort}(m)$, and the corresponding $r_{S1}(r_{L1})$, $r_{S2}(r_{L2})$ and $r_{S3}(r_{L3})$. Note that the independent variable $m$ is not equivalent to the original radius $r$, but only indicates the position. Under various atmosphere and ocean environments in the globe, the values of $r_{S1}(r_{L1})$, $r_{S2}(r_{L2})$ and $r_{S3}(r_{L3})$ vary. Based on the distribution of global $r_{S1}(r_{L1})$, $r_{S2}(r_{L2})$ and $r_{S3}(r_{L3})$ (Fig. 2&Fig. R1), the approximate values in Eqn. 3-5 are obtained. The text is revised to clarify in Line 122-143 (Line 136-162 for text with changes tracked).

*6. Lines 127-128, How can we see that $r_{L3}$ is related to WSP10 from Figure 2c?*

Response: We apologize that the relation cannot be seen directly from Figure 2c. According to Appendix C, we derived local $r_{L3}$ for each grid point, and its global distribution of occurrence frequency in percentage is shown in Figure 2c. We found that unlike the other nodes (Fig. 2a, b, d-f), the distribution of $r_{L3}$ is not concentrated in a single constant, while there is a 92.53% concentration between 55 and 90 $\mu m$. And then we got the relation of $r_{L3}$ (55-90 $\mu m$) and WSP10 (Figure R1). It is seen that $r_{L3}$ is related to WSP10. The figure is added to the supplementary (Fig. S1), and text is revised accordingly in Line 141-143 (Line 160-161 for text with changes tracked).

[Figure]

**Figure R1.** The scatter plot of $r_{L3}$ (y-axis) vs 10-m wind speed (x-axis)

*7. Figure 2 is very difficult for me to understand. As far as I can understand from the text, if 3 GQ nodes are determined, then 3 percentage values are determined. But in Figure 2, the authors use $r_{S1}(r_{L1})$, $r_{S2}(r_{L2})$ and $r_{S3}(r_{L3})$ as the name of the x-axis, and there are so many bars in each panel of Figure 2, what do the bars represent? I strongly recommend that the authors devote more space to the introduce the new methods in their manuscript to respond to my comments 5-7.*

Response: We apologize for the confusion. As we discussed in comment 5, the 3 nodes for $Q_{S\_sort}(m)$ and $Q_{L\_sort}(m)$ are fixed, but the corresponding radius $r$ for $Q_S(r)$ and $Q_L(r)$ changes under various atmosphere and ocean state. To find out the general law of $r_{S1}(r_{L1})$, $r_{S2}(r_{L2})$ and $r_{S3}(r_{L3})$, using global atmosphere and ocean data, we calculate $Q_S(r)$ and $Q_L(r)$ locally, and thereby $r_{S1}(r_{L1})$, $r_{S2}(r_{L2})$ and $r_{S3}(r_{L3})$ at each grid point. The bars in Figure 2 show the occurrence frequency of different values of $r_{S1}(r_{L1})$, $r_{S2}(r_{L2})$ and $r_{S3}(r_{L3})$. For example, in Fig. 2a, more than 99% of local $r_{L1}$ are about 443.914 $\mu m$, therefore we directly set $r_{L1} = 443.914 \mu m$. As suggested, we add the detailed introduction of the new method in section 3.1.

*8. The authors need to briefly describe how the atmospheric and oceanic components of CFSv2.0-WW3 are initialized. I also note that WW3 is not global, then what dataset is used to provide open boundary conditions for WW3?*

Response: The initial fields at 00:00 UTC of the first day in each experiment for the atmospheric and oceanic components of CFSv2.0 were generated by the real time operational Climate Data Assimilation System (Kalnay et al., 1996), downloaded from the CFSv2.0 official website (http://nomads.ncep.noaa.gov/pub/data/nccf/com/cfs/prod). The initial wave fields were generated from 10-day simulation starting from rest in a stand-alone WW3 model, forced by ERA5 10-m winds and ice concentration. The open boundary conditions of the wave component were also obtained by the global simulation of the stand-alone WW3 model. Text is revised accordingly in Line 163-169 (Line 183-190 for text with changes tracked).

*9. The difference between the spray-mediated heat fluxes calculated by A15 and A92 schemes is so significant. Can the authors comment on what cause this large difference? Extrapolation of VS and VL at high wind speeds? or due to the use of single-radius droplets to represent the full-spectrum integral? As Andreas et al (2015) said, VS &VL in A15 are extrapolated at high wind speeds, while*

*SSGF (sea spray generation function) in A92 are deduced. The lack of discussion on the causes of this discrepancy will greatly diminish the importance of this article. I am seriously concerned about this.*

Response: In the study, the wind speeds are generally less than 25 m/s (e.g. Fig. R1 & Fig. R4), while extrapolation is required only for wind speeds >25 m/s. So, the difference in heat fluxes calculated by A15 and A92 is mainly due to the use of single-radius droplets. The discussion is added as suggested in Line 192-195 (Line 215-219 for text with changes tracked).

*10. It would be better to superimpose the Mean Error onto Figure 6c,7c,8c,9c,10c, and 11c.*

Response: The Mean Error (ME) is shown in Fig. R2, which is calculated as ME=$\sum_{i=1}^{n}(\hat{y}_i - y_i)/n$, where $\hat{y}_i$ is simulated value and $y_i$ is OISST/ERA5 data, and n is the total number of grid points. Considering the positive and negative errors (Fig. R2) might cancel out for ME calculation, we also calculate mean absolute errors (MAE=$\sum_{i=1}^{n}|\hat{y}_i - y_i|/n$) in Fig. R3. The MAE is added in Fig. 6c, 8c, 10c-13c of revised text, and it is consistent with the result of RMSE. The ME figure is added to the supplementary (Fig. S5). The text is revised accordingly as well in section 4.2.

[Figure]

**Figure R2**. The Mean Error of SPRAY-A15 (red) and SPRAY-GQ (black) in Jan-Feb, 2017 (a, c, e) and Aug-Sep, 2018 (b, d, f): (a) SST of 0-360°E, 40-75°S; (b) SST of 0-360°E, 20-75°N; (c&d) WSP10 of 0-360°E, 0-60°N; (e&f) SWH of 0-360°E, 0-60°N.

[Figure]

**Figure R3.** The MAE of SPRAY-A15 (red) and SPRAY-GQ (black) in Jan-Feb, 2017 (a, c, e) and Aug-Sep, 2018 (b, d, f): (a) SST of 0-360°E, 40-75°S; (b) SST of 0-360°E, 20-75°N; (c&d) WSP10 of 0-360°E, 0-60°N; (e&f) SWH of 0-360°E, 0-60°N.

*11. According to Andreas et al. (2015), the effect of sea spray become significant at wind speed of 10-13 ms-1. How strong can the simulated WSP10 be? Therefore, I would like to see the global distribution of WSP10 simulated by CFSv2.0-WW3.*

Response: The global distributions of SPRAY-GQ WSP10 in Jan-Feb, 2017 are shown in Fig. R4. At middle and high latitudes, WSP10s can exceed 10 m/s. Overall, the wind speeds in the simulation are in the range of 0-25 m/s. The discussion is added in Line 222-225 (Line 248-252 for text with changes tracked).

[Figure]

**Figure R4**. The WSP10 (m/s) of SPRAY-GQ in Jan-Feb, 2017: (a) the 14th day; (b) the 35th day; (c) the 56th day; (d) the 53-day average, with the first 3-day simulation discarded.

*12. Line 178, what does equivalent neutral wind mean? Do the authors mean that the equivalent neutral winds in OAFLux are larger than those in ERA5?*
Response: We apologize for the confusion. OAFlux only provides neutral wind speeds, calculated from wind stress and the corresponding roughness by assuming air is neutrally stratified. Previous studies (Seethala et al., 2021; Lindemann et al., 2021) indicated the neutral winds from OAFlux are larger than winds given in ERA5. Text is revised to clarify in Line 202-205 (Line 227-230 for text with changes tracked).

*13. Lines 230-231, The expression is not accurate, the reduced wind and weaker mixing can lead to warmer SST?*
Response: The reduced winds weaken the upper ocean mixing, the water becomes more stratified, and then the SST tends to be warmer, and vice versa. Text is revised accordingly in Line 269-271 (Line 304-305 for text with changes tracked).

*14. As we know, satellite scatterometer and altimeter data are usually used to validate WSP10 and SWH for short term weather forecast. I don't know why the authors use ERA5 reanalysis as validation data for seasonal prediction.*
Response: As suggested, we compare our simulation results with the monthly global ocean RSS Satellite Data Products for WSP10 (https://data.remss.com/wind/monthly_1deg/) and the Reprocessed L4 Satellite Measurements for SWH (https://doi.org/10.48670/moi-00177). Due to the spatial and temporal coverage of satellite data, we can only obtain the monthly averaged satellite data for the globe. So we compare the monthly averaged WSP10 and SWH simulations with the corresponding satellite data (e.g., Fig. R5&R6 in Aug-Sep, 2018). The averaged WSP10 and SWH differences compared with satellite data (Fig. R5a&c; Fig. R6a&c) are consistent with those compared with ERA5 (Fig. R5b&d; Fig. R6b&d). Besides, the differences of WSP10s between

ERA5 and the satellite data are always less than 1 m/s and the differences of SWHs are always less than 0.3 m (Fig. R5e&6e). Since ERA5 provides daily data for comparison and the differences between ERA5 and satellite data are small, we use ERA5 for validation and add the following figures in the supplementary for references (Fig. S9-S12). The discussion is added in Line 274-283 (Line 309-320 for text with changes tracked).

[Figure]

**Figure R5**. The average WSP10 (m/s) differences between SPRAY-A15/SPRAY-GQ and satellite data (a/c; SPRAY-A15/SPRAY-GQ minus satellite data), differences between SPRAY-A15/SPRAY-GQ and ERA5 (b/d; SPRAY-A15/SPRAY-GQ minus ERA5), and differences between ERA5 and satellite data (e; ERA5 minus satellite data) in Aug-Sep, 2018. The dotted areas are statistically significant at 95% confidence level.

[Figure]

**Figure R6**. The average SWH (m) differences between SPRAY-A15/SPRAY-GQ and satellite data (a/c; SPRAY-A15/SPRAY-GQ minus satellite data), differences between SPRAY-A15/SPRAY-GQ and ERA5 (b/d; SPRAY-A15/SPRAY-GQ minus ERA5), and differences between ERA5 and satellite data (e; ERA5 minus satellite data) in Aug-Sep, 2018. The dotted areas are statistically significant at 95% confidence level.

*15. Does the introduction of sea spray can improve the simulation of other elements? For example, air temperature and humidity.*

Response: To check, we compare the simulated 2-m air temperature (T02) and specific humidity (SPH) with ERA5 as well. The differences between CTRL (without sea spray effect) and SPRAY-GQ in Jan-Feb, 2017 are shown in Fig. R7.

Compared with CTRL experiment, the introduction of sea spray cannot significantly reduce the

global overall errors of simulations, but it leads to regional improvements (blue in Fig. R7e&f). For example, T02 and SPH in CTRL are underestimated in the Northwest Pacific (blue in Fig. R7a&b), and SPRAY-GQ experiment improves them by increasing temperature and moisture (Fig. R7c-f). The reduced errors are related to relatively large WSP10s over these areas (Fig. R4), since the effects of sea spray become significant at wind speeds larger than 10 m/s. The results of Aug-Sep, 2018 are also compared and added to the supplementary (Fig. S20-S21), and text is revised accordingly in Line 326-336 (Line 369-381 for text with changes tracked).

[Figure]

**Figure R7**. The 53-day average T02 (a, c, e) and SPH (b, d, f) differences between CTRL and ERA5 (a&b; CTRL minus ERA5), differences between SPRAY-GQ and CTRL (c&d; SPRAY-GQ minus CTRL), and MAE differences between SPRAY-GQ and CTRL (e&f) in Jan-Feb, 2017. The first 3-day simulation is discarded. The dotted areas are statistically significant at 95% confidence level.

*16. Line 246, there is a grammatical mistake in this sentence. And Fig10b&11b do not support 'The SWHs in SPRAY-GQ improve compared with those in SPRAY-A15'. Do the authors mean that there is a significant difference between SPRAY-GQ and SPRAY-A15?*
Response: Yes, thanks. The text is corrected to "The SWHs in SPRAY-GQ are significantly different with those in SPRAY-A15 (Fig. 12b&13b)" in Line 296-297 (Line 334-336 for text with changes tracked).

*17. As far as I know, the sea spray algorithm codes of Andreas are open source, please upload the author's modified A92 codes to a repository so that others can repeat the author's results.*

Response: The codes of our modified A92 (SPRAY-GQ) can be found in https://zenodo.org/record/7100345#.Y66vRtVByHt.

*18. Lines 255-259, the authors try to discuss the physical mechanism that responsible for the accelerated surface wind. However, the citation does not seem to support the author's conclusions. I can understand that the increase of air-sea heat flux could promote air convection in the vertical, but how does it promote the downward transmission of momentum from the upper layer of atmosphere? By affecting the large-scale atmospheric circulation? Please provide appropriate citations or give your own analysis to support your points.*

Response: We apologize for the unclear writing. The air-sea heat flux influences the sea level pressure (SLP) distribution, and thus influences surface winds. For example, compared with SPRAY-A15, the decreased heat flux of SPRAY-GQ in the Northwest Pacific in Aug-Sep, 2018 leads to higher SLP and smaller pressure gradient (Fig. R8), and thus decreased WSP10; while the increased heat flux in the Gulf of Alaska leads to lower SLP and larger pressure gradient (Fig. R8), and thus enhanced WSP10. Text is revised to clarify in Line 305-311 (Line 344-353 for text with changes tracked).

[Figure]

**Figure R8**. The 53-day average wind (m/s) and sea level pressure (hPa) of SPRAY-A15 (a) and

SPRAY-GQ (b), and their differences (c; SPRAY-GQ minus SPRAY-A15) in Aug-Sep, 2018.

*19. Lines 263-264, again, without verification by direct/indirect observation, we can't say that SPRAY-GQ is more accurate than A15. All we can say is that the difference between SPRAY-GQ and A92 is smaller than that between A15 and A92.*

Response: We agree. The text is revised as suggested in Line 314-316 (Line 356-359 for text with changes tracked).

*20. Finally, I think the current experimental design is insufficient, a reference experiment without sea spray effect is missing in manuscript. Although it may be expensive to conduct a new set of experiment, it makes sense for the scientific community to understand the importance of spray-mediated heat flux for seasonal and intra-seasonal prediction.*

Response: As suggested, we add a new experiment without sea spray effects (CTRL). The introduction of sea spray cannot significantly reduce the global overall errors of simulations, but it leads to regional improvements (e.g., Fig. R7, Fig. R9-11). For example, compared with CTRL in Jan-Feb, 2017, SST MAE of SPRAY-GQ in the southeast of Australia decreases (blue in Fig. R9e), because of warmer SST (Fig. R9c) related to reduced wind (Fig. R10c). The reduced wind here also leads to lower SWH (Fig. R11c) and thus reduced SWH overestimation (Fig. R11e). The related content is added in Line 326-336 (Line 369-381 for text with changes tracked) and the following figures are added in the supplementary for references (Fig. S17-S21).

[Figure]

**Figure R9**. The 53-day average SST differences between CTRL and OISST (a&b; CTRL minus OISST), differences between SPRAY-GQ and CTRL (c&d; SPRAY-GQ minus CTRL), and MAE differences between SPRAY-GQ and CTRL (e&f) in Jan-Feb, 2017 (a, c, e) and in Aug-Sep, 2018

[Figure]

**Figure R10**. The 53-day average WSP10 differences between CTRL and ERA5 (a&b; CTRL minus ERA5), differences between SPRAY-GQ and CTRL (c&d; SPRAY-GQ minus CTRL), and MAE differences between SPRAY-GQ and CTRL (e&f) in Jan-Feb, 2017 (a, c, e) and in Aug-Sep, 2018 (b, d, f). The first 3-day simulation is discarded. The dotted areas are statistically significant at 95% confidence level.

[Figure]

**Figure R11**. The 53-day average SWH differences between CTRL and ERA5 (a&b; CTRL minus ERA5), differences between SPRAY-GQ and CTRL (c&d; SPRAY-GQ minus CTRL), and MAE differences between SPRAY-GQ and CTRL (e&f) in Jan-Feb, 2017 (a, c, e) and in Aug-Sep, 2018 (b, d, f). The first 3-day simulation is discarded. The dotted areas are statistically significant at 95% confidence level.

References

Andreas, E. L., and Emanuel, K. A.: Effects of sea spray on tropical cyclone intensity, Journal of the atmospheric sciences, 58, 3741-3751, 2001.

McClarren, R.: Gauss Quadrature and Multi-dimensional Integrals, Computational Nuclear Engineering and Radiological Science Using Python; Academic Press: Cambridge, MA, USA, 287-299, 2018.

Kalnay, E., Kanamitsu, M., Kistler, R., Collins, W. D., Deaven, D. G., Gandin, L. S., Iredell, M. D., Saha, S., White, G. H., and Woollen, J.: The NCEP/NCAR 40-Year Reanalysis Project, Bulletin of the American Meteorological Society, 77, 437-471, http://dx.doi.org/10.1175/1520-0477(1996)077%3C0437:TNYRP%3E2.0.CO;2, 1996.

Andreas, E. L., Mahrt, L., and Vickers, D.: An improved bulk air–sea surface flux algorithm, including spray-mediated transfer, Quarterly Journal of the Royal Meteorological Society, 141, 642-654, 2015.

Lindemann, D., Avila-Diaz, A., Pezzi, L., Rodrigues, J., Freitas, R. A., Coelho, L., Alonso, M., and Cerón, W. L.: The Surface Wind Influence on the Heat Fluxes Variability on the South Atlantic, 2021.

Seethala, C., Zuidema, P., Edson, J., Brunke, M., Chen, G., Li, X. Y., Painemal, D., Robinson, C., Shingler, T., and Shook, M.: On assessing ERA5 and MERRA2 representations of cold-air outbreaks across the Gulf Stream, Geophysical research letters, 48, e2021GL094364, 2021.

We sincerely appreciate the reviewer for her/his constructive comments on the manuscript. Our responses are listed as follows in blue. Text is revised accordingly.

**Review from Referee #2**

*The authors present a new numerical method to calculate sea spray induced heat fluxes given a sea spray generation function. Specifically, the authors propose the Gaussian Quadrature (GQ) method as a computationally efficient alternative to the spectrum integral method, and more accurate method than simplifications of the integral method. The results show improved agreement of the GQ method with the spectrum integral in comparison to the simplified parameterization. Impact of such difference in the context of world's oceans are presented using a coupled ocean-atmospheric-wave model.*

*Overall, the manuscript is interesting and the modelling community may benefit significantly from the proposed GQ-method. However, the authors are not clear in the type of advancement and implications their work has on current and future works. The method does not improve the physics of sea spray induced heat fluxes, instead it improves the numerical accuracy of solving the involved equations. While I nevertheless believe this is important work worthy to be published, it is an important distinction that is currently not appropriately phrased nor discussed. I therefore recommend a major revision.*

*Major concerns*

*As per the above, I believe that the authors need to be more explicit in what their method is improving. I think that the authors haven't necessarily overstated their conclusions, but the interpretations of some statements are right now too ambiguous or sometimes incorrect. This largely involves the word 'improved' throughout the manuscript, including the title and abstract, where it reads as if the parameterizations are improved. This is not really true, it is actually the numerical method used to solve the parameterizations that has been improved (at least against the simplified parameterization discussed in the manuscript). Thus, when these different methods are compared when implemented in the applied coupled model, the authors are not improving the bulk parameterizations, but simply presenting the numerical error of the methods discussed in the context of the coupled model. Importantly, it is the numerical error assuming that the sea spray generation function is correct.*

Response: As suggested, we revise the title, the abstract and the text to focus on the improvement of the numerical method. The title is revised to "Accelerated Estimation of Sea Spray-Mediated Heat Flux Using Gaussian Quadrature: Case Studies with a Coupled CFSv2.0-WW3 System". For the result comparison in the context of the coupled system, we also clarify that we are not improving the bulk parameterization, but presenting the numerical errors of the methods in Line 219-220 (Line 244-245 for text with changes tracked).

*A related point is the absence of a clear discussion on the interpretation of the results. As mentioned by the authors, the sea spray generation function has an uncertainty of several orders of magnitude. Thus, if the RMSE of the proposed GQ method reduces the numerical error by say 1-10 $W/m^2$ (e.g.,*

*figs 3-5), how relevant is such an improvement in the broader context of the physics and model uncertainty? Such uncertainty in the physics could perhaps get into the 100s of W/m². I'm not suggesting the GQ method proposed here is therefore irrelevant, but it does change the interpretation and application of the model/results in practice, for now and the future. This also brings up some other questions regarding the interpretation of the improved SST as observed in Figs 6 and 7. The uncertainty in the physics of sea spray is considerable larger than the improvement in approximating the spectral integral using the GQ method. Thus, any improvements in the modelled SST cannot be reliably be assigned to the usage of the GQ method.*

Response: We apologize for the unclear writing. Although the sea spray generation function (SSGF) has an uncertainty of several orders of magnitude, the sea spray-mediated heat fluxes in A92 have been tuned by non-negative constants based on observations and the COARE algorithm to reduce the uncertainties (Andreas and Decosmo, 2002; Andreas et al., 2008; Andreas et al., 2015; Andreas, 2003). In this study, we use the constants (Eqn. A7-A8 in Appendix A) for the SSGF (Fairall et al., 1994) to get a mean bias of 3.70 and 0.095 W/m² for latent and sensible heat flux respectively in A92 compared to observations (Andreas et al., 2015). Therefore, a few W/m² improvements of numerical errors in this study are relevant. Even though, we agree that the improved SST and other variables cannot be reliably assigned to the usage of the GQ method, due to the uncertainties of the coupled model itself and SSGF. A discussion about the uncertainty as suggested is added to clarify in Line 350-354 (Line 396-409 for text with changes tracked).

*Minor comments:*

*Line 66: 'huge amount' sounds a bit vague. Especially later on the authors actually provide a number, so they can be more accurate here.*

Response: To be more accurate, we calculate the runtime of CFSv2.0-WW3 global experiments for 7-day forecast with different parameterizations (Table R1). The text is revised as suggested in Line 70-72 (Line 77-79 for text with changes tracked).

**Table R1.** The runtime of CFSv2.0-WW3 global experiments for 7-day forecast with different parameterizations.

| 7-day Forecast | Runtime (h) |
|---|---|
| SPRAY-A92 | 126.94 |
| SPRAY-A15 | 7.60 |
| SPRAY-GQ | 7.67 |

*Line 70: 'apt to produce significant bias', more a numerical error. However, as per one of my major concerns, is this bias/error of significance in the context of the existing uncertainty in sea spray parameterizations?*

Response: Considered that the A92 has a mean bias of 3.70 and 0.095 W/m² for latent and sensible heat flux compared to observations (Andreas et al., 2015), the numerical error here is significant. Text is revised in Line 75 (Line 83 for text with changes tracked).

*Line 86: 'thus provided reliable hourly estimates', is very vague. Either provide a reference about its reliability, or simply state that this dataset is used.*

Response: As suggested, the text is revised to state that this dataset is used in Line 90-93 (Line 99-103 for text with changes tracked).

*Line 100: This sentence reads a bit odd. 'based on eddy correlation observations' refers to the turbulent heat fluxes in cited papers, not the sea spray induced heat flux.*

Response: We agree. The text is revised to "Based on observations of total turbulent heat fluxes and the COARE algorithm" in Line 107-108 (Line 118-119 for text with changes tracked).

*Line 108: 'requires huge amount', just say is computationally expensive.*

Response: Corrected in Line 115 (Line 129 for text with changes tracked). Thanks.

*Line 123-124: somewhat confusing, please rephrase sentence.*

Response: The text is revised to "Since the sea spray-mediated heat flux is not sensitive to salinity (Fig. 1e&f) and only monthly observational data is available, the ESA monthly salinity is applied" in Line 136-137 (Line 151-153 for text with changes tracked).

*166: '36 times' this is an interesting and useful statistic. Perhaps consider to provide what this means in terms of the fully coupled model run, i.e., is it still saving much time percentage wise?*

Response: As suggested, we test the runtime of the fully coupled experiments for 7-day forecast (Table R1). The runtime of SPRAY-GQ experiment is about 17 times less than the runtime of SPRAY-A92 experiment. Text is revised accordingly in Line 215-218 (Line 240-243 for text with changes tracked).

*Line 201-202: confusing sentence, please rephrase.*

Response: The text is revised to "The increased (decreased) SSTs north (south) of 50°S in SPRAY-GQ compared to those in SPRAY-A15 (Fig. 6b) reduce the RMSE of SST in SPRAY-GQ" in Line 235-237 (Line 266-268 for text with changes tracked).

*Line 204: 'fig 12g' best practice is to keep order of figures intact.*

Response: Corrected in Line 242 (Line 275 for text with changes tracked). Thanks.

*Line 233: 'significant improvements', I disagree. Fig. 8c and 9c seem to show very similar RMSE. Their variability after day 18 is very similar.*

Response: The sentence is revised in Line 273 (Line 307 for text with changes tracked). The differences of WSP10 RMSEs between SPRAY-GQ (black) and SPRAY-A15 (red) are very small in the first two weeks. Afterwards, the mean RMSE in SPRAY-GQ is lower than that in SPRAY-A15 significantly at 95% confidence level in both boreal winter (Fig. 10c in revised text) and boreal summer (Fig. 11c in revised text).

*Line 260: This manuscript would greatly benefit from a separate and in-depth discussion on the interpretation and implications of the results.*

Response: Thanks. The following discussion is added in Line 355-364 (Line 410-420 for text with

changes tracked).

When wind speed is larger than 10 m/s, spray-mediated heat flux can become as important as the interfacial heat flux (Andreas and Decosmo, 1999, 2002). Particularly, even in the absence of air-sea temperature difference, the spray-mediated sensible heat flux is still present (Andreas et al., 2008). As indicated by previous studies (e.g., Garg et al., 2018; Song et al. 2022), it is necessary to superimpose the spray-mediated heat flux on the bulk formula to complete the physics of turbulent heat transfer for coupled simulation. Since the full microphysical parameterization (A92) is computationally expensive, an efficient algorithm that captures the main features of A92 can be beneficial to large-scale climate systems or operational storm models. The GQ method proposed in the study can efficiently calculate the spray-mediated heat flux, and agree better with A92 than A15. Thereby, the GQ based spray-mediated heat flux is promising to be widely applied in large-scale climate systems and operational storm models.

References

Andreas, E. L., and Decosmo, J.: The signature of sea spray in the HEXOS turbulent heat flux data, Boundary-layer meteorology, 103, 303-333, 2002.

Andreas, E. L.: 3.4 AN ALGORITHM TO PREDICT THE TURBULENT AIR-SEA FLUXES IN HIGH-WIND, SPRAY CONDITIONS, 2003.

Andreas, E. L., Persson, P. O. G., and Hare, J. E.: A bulk turbulent air–sea flux algorithm for high-wind, spray conditions, Journal of Physical Oceanography, 38, 1581-1596, 2008.

Andreas, E. L., Mahrt, L., and Vickers, D.: An improved bulk air–sea surface flux algorithm, including spray-mediated transfer, Quarterly Journal of the Royal Meteorological Society, 141, 642-654, 2015.

Fairall, C., Kepert, J., and Holland, G.: The effect of sea spray on surface energy transports over the ocean, Global Atmos. Ocean Syst, 2, 121-142, 1994.

Andreas, E. L., and Decosmo, J.: Sea spray production and influence on air-sea heat and moisture fluxes over the open ocean, in: Air-sea exchange: physics, chemistry and dynamics, Springer, 327-362, 1999.

Song, Y., Qiao, F., Liu, J., Shu, Q., Bao, Y., Wei, M., and Song, Z.: Effects of sea spray on large-scale climatic features over the Southern Ocean, Journal of Climate, 1-51, 2022.

Garg, N., Ng, E. Y. K., and Narasimalu, S.: The effects of sea spray and atmosphere–wave coupling on air–sea exchange during a tropical cyclone, Atmospheric Chemistry and Physics, 18, 6001-6021, 2018.

---

## Author Response (AR3)

We sincerely appreciate the reviewer for her/his constructive comments on the manuscript. Our responses are listed as follows in blue. Text is revised accordingly.

**Review from Referee #1**

*Line 66, to my knowledge, this point is still under debate or even out of date if you see Smith et al. (2014). Perhaps the author should rephrase your sentence.*
*Smith, R.K., Montgomery, M.T. and Thomsen, G.L. (2014), Sensitivity of tropical-cyclone models to the surface drag coefficient in different boundary-layer schemes. Q.J.R. Meteorol. Soc., 140: 792-804. https://doi.org/10.1002/qj.2057*
Response: We agree that it is under debate how the variations in surface drag coefficients influence tropical cyclone intensity (e.g., Emanuel, 1995; Smith et al., 2014). As Smith et al. (2014) indicated, the increase of the inflow induced by enhanced surface drag coefficient tends to spin up the tropical cyclone, and thus might offset the frictional torque to intensify the cyclone. The discussion of drag coefficients is beyond the scope of our study. To avoid confusion, we delete the misleading sentences in Line 60 (Line 60-67 for text with changes tracked).

Emanuel, K. A.: Sensitivity of tropical cyclones to surface exchange coefficients and a revised steady-state model incorporating eye dynamics, Journal of Atmospheric Sciences, 52, 3969-3976, 1995.
Smith, R. K., Montgomery, M. T., and Thomsen, G. L.: Sensitivity of tropical‐cyclone models to the surface drag coefficient in different boundary‐layer schemes, Quarterly Journal of the Royal Meteorological Society, 140, 792-804, 2014.

We sincerely appreciate the reviewer for her/his constructive comments on the manuscript. Our responses are listed as follows in blue. Text is revised accordingly.

**Review from Referee #2**

*Overall I think the manuscript is near-acceptable, with the exception of the abstract. While most of my comments have, in my opinion, been properly addressed, I think this is not the case for lines 20-24 in the abstract (from For experiments ... that of A15). To me it still reads here as if SPRAY-GQ is improving simulations of sea surface temperature and other properties without providing the clarity as to why this is. That is, the physics are not improved, simply the numerical error is reduced. While this is quite clear now in the manuscript, in the abstract, this needs to be further clarified to make sure words like 'significantly improved' are interpreted correctly by the reader.*
Response: As suggested, we added the sentence "These improvements are due to the reduced numerical errors" to clarify in Line 23-24.

*Please also have a re-read of the manuscript for grammar. Some errors for example:*
*Line 407: even though --> However*
*Line 418: thereby --> therefore*
Response: Thanks. We have checked the manuscript for grammar and revised as suggested.